# Investigation of Deficits in Auditory Emotional Content Recognition by Adult Cochlear Implant Users through the Study of Electroencephalographic Gamma and Alpha Asymmetry and Alexithymia Assessment

**DOI:** 10.3390/brainsci14090927

**Published:** 2024-09-17

**Authors:** Giulia Cartocci, Bianca Maria Serena Inguscio, Andrea Giorgi, Dario Rossi, Walter Di Nardo, Tiziana Di Cesare, Carlo Antonio Leone, Rosa Grassia, Francesco Galletti, Francesco Ciodaro, Cosimo Galletti, Roberto Albera, Andrea Canale, Fabio Babiloni

**Affiliations:** 1Department of Molecular Medicine, Sapienza University of Rome, Viale Regina Elena 291, 00161 Rome, Italy; dario.rossi@uniroma1.it (D.R.); fabio.babiloni@uniroma1.it (F.B.); 2BrainSigns Ltd., Via Tirso 14, 00198 Rome, Italy; 3Department of Computer, Control, and Management Engineering “Antonio Ruberti”, Sapienza University of Rome, Piazzale Aldo Moro 5, 00185 Rome, Italy; biancams.inguscio@uniroma1.it; 4Department of Anatomical, Histological, Forensic & Orthopedic Sciences, Sapienza University of Rome, Piazzale Aldo Moro 5, 00185 Rome, Italy; andrea.giorgi@uniroma1.it; 5Institute of Otorhinolaryngology, Catholic University of Sacred Heart, Fondazione Policlinico “A Gemelli”, IRCCS, Largo Agostino Gemelli 8, 00168 Rome, Italy; walter.dinardo@unicatt.it (W.D.N.); tizianadicesare90@gmail.com (T.D.C.); 6Department of Otolaringology Head-Neck Surgery, Monaldi Hospital, Via Leonardo Bianchi, 80131 Naples, Italy; carloantonioleone@hotmail.com (C.A.L.); rosa.grassia@ospedalideicolli.it (R.G.); 7Department of Otorhinolaryngology, University of Messina, Piazza Pugliatti 1, 98122 Messina, Italy; fgalletti@unime.it (F.G.); francesco.ciodaro@unime.it (F.C.); cosimogalletti92@gmail.com (C.G.); 8Department of Surgical Sciences, University of Turin, Via Genova 3, 10126 Turin, Italy; roberto.albera@unito.it (R.A.); andrea.canale@unito.it (A.C.); 9Department of Computer Science, Hangzhou Dianzi University, Hangzhou 310018, China

**Keywords:** emotion, EEG, gamma, alpha, asymmetry, music, deafness, sensorineural hearing loss, cochlear implant, alexithymia

## Abstract

Background/Objectives: Given the importance of emotion recognition for communication purposes, and the impairment for such skill in CI users despite impressive language performances, the aim of the present study was to investigate the neural correlates of emotion recognition skills, apart from language, in adult unilateral CI (UCI) users during a music in noise (happy/sad) recognition task. Furthermore, asymmetry was investigated through electroencephalographic (EEG) rhythm, given the traditional concept of hemispheric lateralization for emotional processing, and the intrinsic asymmetry due to the clinical UCI condition. Methods: Twenty adult UCI users and eight normal hearing (NH) controls were recruited. EEG gamma and alpha band power was assessed as there is evidence of a relationship between gamma and emotional response and between alpha asymmetry and tendency to approach or withdraw from stimuli. The TAS-20 questionnaire (alexithymia) was completed by the participants. Results: The results showed no effect of background noise, while supporting that gamma activity related to emotion processing shows alterations in the UCI group compared to the NH group, and that these alterations are also modulated by the etiology of deafness. In particular, relative higher gamma activity in the CI side corresponds to positive processes, correlated with higher emotion recognition abilities, whereas gamma activity in the non-CI side may be related to positive processes inversely correlated with alexithymia and also inversely correlated with age; a correlation between TAS-20 scores and age was found only in the NH group. Conclusions: EEG gamma activity appears to be fundamental to the processing of the emotional aspect of music and also to the psychocognitive emotion-related component in adults with CI.

## 1. Introduction

Emotions represent a key item in communication purposes. In fact, from infancy, the processing of emotional expressions would develop into the capability of recognizing different emotions by leveraging personal experience and through the maturation of sensory and perceptual systems, particularly auditory and visual ones. In this framework, sensory deprivation, for instance, caused by deafness, would negatively affect such skills acquisition [1,2]. Sensorineural hearing loss can be caused by many different etiologies, and the assessment if the hearing loss is congenital or delayed and if its origin is genetic or nongenetic appears extremely relevant [3]. This condition corresponds to a wide range of congenital or acquired causes that may occur in the pre-, peri-, or postlingual stages of language development. Many authors suggested that cochlear implants (CIs) represent a suitable treatment for children and adult patients with severe (hearing loss of 71–90 dB HL) to profound (hearing loss over 90 dB HL) in the conversational frequency range (from 500 to 4000 Hz), where no or minimum benefit from a hearing aid after a trial period of 3–6 months was obtained, according to NICE Guidance on cochlear implantation (NICE 2009). For a more thorough description of the indications for CIs in accordance with age and hearing loss characteristics in both ears, see, e.g., [4]. The causes of hearing loss in childhood (excluding infectious pathology of the middle ear) may be extrinsic (embryofoetopathy, meningitis, trauma, drug ototoxicity, noise trauma, etc.), genetic (e.g., alterations at the DFNB1 locus, STRC pathogenic variations or alterations at the DFN16 locus, SLC26A4 pathogenic variations, OTOF pathogenic variations, POU3F4 pathogenic variations or alterations at the DFNX2 locus), or both. The prevalence of genetic sensorineural hearing impairment is currently estimated to be 66% in industrialized countries, and hereditary hearing loss can be divided into two broad categories: nonsyndromic, estimated at 70–90%, and syndromic, estimated at 10–30% [5]. The time of deafness acquisition can be divided into prelingual (onset of deafness before 2 years of age), perilingual (onset of deafness at 2–3 years of age), and postlingual (onset of deafness after 4 years of age). It is obvious that the development and functioning of the auditory system in congenitally deaf children who have recovered their hearing through CIs is not physiological [6,7,8,9], but it is not clear whether such alterations are also extended to adult CI users that often acquire deafness later in life. Thanks to technological developments, CI users (especially postlingually deaf ones) are able to achieve excellent speech recognition in quiet environments, but there are still limitations in the ability of CI sound processors to provide fine spectrotemporal information, resulting in difficulties for CI users to perceive complex acoustic cues such as music, environmental sounds, lexical tones, and voice emotion [10].

Emotion recognition deficits in CI users have been extensively studied, mostly through the evaluation of performance and ratings, through the assessment of prosody recognition [11,12,13], and, interestingly, by focusing not only on the perception but also on the expression of emotional vocal cues [14]. Despite clear deficits in emotion recognition in CI users compared to normal hearing (NH) controls, both groups presented age-related deficits in emotion identification (considering both accuracy and sensitivity) [15]. Interestingly, another similarity between CI users and NH occurs in emotion recognition tasks characterized by incongruent prosodic and lexical–semantic cues: CI users rely more on lexical–semantic cues and the same NH controls when exposed to spectrally degraded stimuli [16]. In this framework, the comparison between bimodal and bilateral CI users was also performed, reporting higher (but not reaching the statistical significance) performances of bimodal users compared to bilateral CI users, on four pitch-related tests (hearing in noise test, Montreal battery of evaluation of amusia, aprosodia battery, talker identification using vowels), interestingly, suggesting that dealing with these “real world” tasks is not only a matter of pitch discrimination but requires additional skills [17]. Moreover, alterations in prosody were identified not only in speech perception but also in speech production by CI users, showing that duration and rhythm features were altered between CI users and controls [18].

Previous behavioural studies on adult CI users in a musical excerpt emotion recognition task showed that although CI users performed above chance level, their correct responses were impaired for three of the four emotions tested (happy, scary, and sad, but not for peaceful excerpts), also highlighting deficits in the perception of arousal of musical excerpts, whereas valence rating remained unaffected [19]. Such evidence has been linked to a selective preservation of timing and rhythmic musical cues, but not of the perception of spectral (pitch and timbre) musical dimensions in adult CI users [20,21,22,23]. An interesting contemporary study assessed pitch discrimination, music processing, and speech intelligibility in CI users and NH controls, obtaining that CI users performed worse than NH controls on all the tasks, and suggesting that also in correspondence of musical emotional stimuli CI users relied more on processes underlying speech intelligibility [24]. Concerning the kinds of emotion employed in the present study, that is, happy and sad [25], it has already been proven from a behavioural perspective that CI users are able to discriminate happy from sad auditory emotions, but with lower performances than NH controls [26,27]. Similarly to the already mentioned study performed on adult CI users assessing both emotional music and speech categorization skills [24], such evaluation has been made also on children CI users, contrary to adults showing a correlation between performance on the speech task with performance on the music task, and implant experience was correlated with performance on both tasks [28]. 

In the context of emotional functioning, previous research has highlighted the importance of emotion regulation and alexithymia. Emotion regulation refers to the conscious and unconscious strategies used in response to specific emotional experiences [29], while alexithymia [30] is characterized by the difficulty of verbalizing emotions and developing fantasies [31]. However, despite studies on the mental health of adults with hearing impairment, and studies focusing on the delicate period of preadolescence and adolescence on emotional abilities (e.g., [32]), there is little research on the emotional functioning of these individuals. One of these few studies on adult deaf participants included both hearing aids and CI users, focused on the investigation of wellbeing in deaf people, and identified differences between deaf and NH persons in terms of alexithymia, despite authors reporting no differences between NH and deaf persons concerning emotional functioning [33]. From the same group, a recent study showed that deaf participants reported increased alexithymia scores and lower scores for positive relationships, as respectively evaluated through the Toronto Alexithymia Scale (TAS-20 [34]) and the Psychological Wellbeing Scale (PWBS [35]), whilst no differences were found between deaf and NH participants on emotional regulation as indexed by the Trait Meta-Mood Scale (TMMS-24 [36]) [37].

Concerning neural signatures of emotion processing, previous studies have shown that brain activity in the electroencephalographic (EEG) gamma frequency range is sensitive to emotion recognition [38,39]. Moreover, right brain dominance in gamma activity has been associated with emotional processing of faces compared to neutral faces [40]. Furthermore, differences in gamma activity between cochlear implant (CI) users and individuals with normal hearing (NH) have been shown both when (i) employing verbal stimuli, with NH users presenting higher gamma activity than CI users, irrespectively of the emotional valence [41]; (ii) employing nonverbal emotional vocalizations, evidencing relative higher right hemisphere gamma activity in the UCI children group in comparison to the NH group [42]. This suggests that gamma brain activity may influence emotion recognition depending on the type of stimuli received. Furthermore, concerning EEG gamma band and music, a relation was found between individual gamma frequency and music perception (for a review [43]) and, in particular, research investigating musical expertise in children showed that gamma-band activity in musically trained children was higher over frontal areas, compared to children without musical training [44]. Concerning EEG alpha asymmetry and music perception in adult and children samples, it was reported that unilateral CI (UCI) users presented altered alpha asymmetry patterns in comparison to bilateral CI users (BCI), and that BCI presented alpha asymmetry patterns more similar to NH controls in all the included conditions (original, distorted, and mute audio–video) [45].

Given the importance of emotion recognition for communication purposes, and the impairments in CI users for such skill, despite technological improvements enabling impressive language recognition performances, the aim of the present study was the investigation of the neural correlates of emotion recognition skills apart from language in adult unilateral CI (UCI) users during a music in noise recognition task (happy/sad categorization task). Importantly, such eventual alterations have been investigated through a neuroscientific approach and not only from a behavioural perspective given the evidence of a further informativity provided by such methods [46,47]. Finally, in order to assess the eventual comorbidity of deafness with specific psychological traits linked to emotion processing, the TAS-20 [48,49,50,51] for measuring alexithymia has been administered to participants.

## 2. Materials and Methods

### 2.1. Sample

A total of 20 adult unilateral cochlear implant users (UCI), 12 left and 8 right implanted side (11F, 9M; mean age ± st. dev.: 45.90 ± 12.73), and 8 normal hearing (NH) controls (5F, 3M; mean age ± st. dev.: 38.75 ± 12.21) were included in the study. The NH and UCI groups were not statistically different concerning age (t = −1.357 *p* = 0.186) and sex (Fisher’s exact test two-tailed *p* = 1). Participants were all right-handed. Concerning the etiology of deafness, 11 were postlingual, 3 were perilingual, and 6 were prelingual deaf patients. During the test, none of the CI users wore hearing aids in their contralateral ear. The audiometric inclusion criterion for UCI was a word comprehension rate of rate of at least 50% at 65 dB SPL [52], and this intensity was used for stimulus delivery in the experiment. The 50% threshold was set because a common measure of a listener’s ability to understand speech in noise is the speech reception threshold (SRT) [53], which is defined as the SNR at which 50% of speech is correctly understood. Concerning the UCI group, the degree of hearing loss in the contralateral ear to the implanted one presented an average PTA (calculated including the frequencies 250–500–1000–2000–4000 Hz) ± standard deviation: 101.05 ± 13.77 dB. Furthermore, in the UCI group, the age at implantation was 40.40 ± 14.59 years old, the duration of deafness (from the onset of deafness to implantation date) was 25.25 ± 14.64 years, the period of CI use at the time of the experiment was 5.16 ± 6.17 years, and all patients used hearing aids prior to CI use, according to Italian guidelines for cochlear implantation (https://www.iss.it/-/impianto-cocleare-adulto-bambino, accessed on 27 August 2024). The UCI group was also divided into two subgroups on the basis of the deafness etiology: pre/perilingual (6F, 3M; mean age ± st. dev.: 37.44 ± 12.85) and postlingual group (5F, 6M; mean age ± st. dev.: 52.82 ± 7.68), of course presenting a statistically significant increase concerning age for the postlingual group in comparison to the prelingual one (t = −3.321 *p* = 0.004), justified by the fact that typically, in Italy, it is more frequent to have younger pre-/perilingual adult CI users thanks to the quite recent implementation and improvement over all the Italian territory of the national neonatal screening program that allows access to early auditory rehabilitation to younger patients [54]. In accordance with this, and with the update of the guidelines for the definition of CI candidates, it is, in fact, expected to see a progressive decrease in the proportion of candidates for a CI with long durations of profound deafness, because patients undergo CI surgery earlier in the time course of their deafness [55].

### 2.2. Protocol

Participants were familiarized with the protocol using musical stimuli that were not used in the real study, but were part of the same database as the experimental stimuli. Participants, equipped with the EEG cap, were sitting in front of a computer, instructed to listen to the musical stimuli and to limit movements as much as possible. Each musical stimulus was preceded by a white screen (1500 ms) and a gray screen with a fixation cross (3000 ms). After each musical excerpt, a happy and a sad face drawings, already used in other studies using the same music database [56,57], appeared simultaneously on the screen. Then, using a customized keyboard, participants were instructed to press the button below each face to assign the presumed emotional content to each excerpt. Half of the correct responses corresponded to the right button and half to the left button, displayed in a randomized order among participants. There was no time limit for giving any responses. Participants were instructed to use their favorite hand for giving responses either through the keyboard buttons and the touchpad and after giving each response to return to the same rest position, in order to homogenize the starting position of each trial and subtrial phase. 

The study was carefully explained to all participants, and it was made clear that they would not receive any form of compensation for their participation and that they could withdraw from the experiment at any time without having to give an explanation. Participants, after being allowed to ask any questions they might have to the experimenter, signed an informed consent to their participation. The study was approved by the Gemelli Hospital Ethical Committee, and was conducted according to the principles outlined in the Declaration of Helsinki of 1975, as revised in 2000.

### 2.3. Stimuli

The stimuli were excerpts of classical piano music, previously categorized as happy (e.g., Beethoven’s Symphony No. 6) or sad (e.g., Albinoni’s Adagio) in a database made available by the authors [25] and already employed in studies concerning the recognition of musical features by CI users [26,56]. From the original database of 32 musical excerpts, 2 lists of 24 items each were generated after a pretest on the recognition of the emotional content of the excerpts by NH university students (*n* = 32), in order to exclude the excerpts characterized by the poorest recognition, with an average duration of 15.386 ± 5.617 s per excerpt. Each list included 24 musical stimuli, composed of 8 musical excerpts, half happy and half sad, belonging to the original database, delivered in quiet (Q), and at two signal-to-noise ratios (SNRs): 10 and 5 [58]. The employed background noise for SNR conditions was continuous 4-talker babble background noise [59], with SNR5 being the most difficult audibility condition among those presented. Stimuli were delivered free-field through two loudspeakers placed in front of and behind the participant at the distance of 1 m each [42], so as to meet CIs best requirements for their use, at an average intensity of 65 dB SPL, measured at the participant’s head [26,60]. None of the participants was a musician or was affected by psychiatric or neural diseases, nor used drugs with psychoactive effect at least in the six months preceding the experiment. UCI participants were also asked about their habits concerning mean hours of listening to music per week [61], resulting in the majority of the sample (55.56%) as reserving 0–2 h per week for such activity; the least percentage of responders (both 7.41%) reserved 3–4 h/week and >9 h/week obtained and 5–6 h/week for 11.11% of the responders; 18.51% of the UCI participants preferred to not respond to such a question.

Stimuli were delivered in a randomized order through E-prime 3.0 software (Psychology Software Tools, Inc., Pittsburgh, PA, USA), that allowed also the collection of behavioural data.

### 2.4. EEG 

A digital EEG system (Beplus EBNeuro, Florence, Italy) was used to record 20 EEG channels (Fpz, Fz, F3, F4, F7, F8, Cz, C3, C4, Cp5, Cp6, T7, T8, Pz, P3, P4, P7, P8, O1, O2) according to the international 10/20 system, with a sampling frequency of 256 Hz. The impedances were maintained below 10 kΩ, and a 50 Hz notch filter was applied to remove the power interference. A ground electrode was placed on the forehead and reference electrodes on earlobes. The EEG signal was initially band-pass filtered with a 5th-order Butterworth filter (high-pass filter: cut-off frequency fc = 1 Hz; low-pass filter: cut-off frequency fc = 40 Hz). Through the application of a regression-based method, eyeblinks artifacts were identified and corrected. In particular, the Fpz channel was used to identify and remove eye-blink artifacts by the use of the REBLINCA algorithm [62]. For other sources of artifacts (e.g., environmental noise, user movements, etc.), specific procedures of the EEGLAB toolbox were employed [63]. In particular, the EEG dataset was firstly segmented into epochs of 2 s through moving windows shifted by 0.125 s. This windowing was chosen with the compromise of having both a high number of observations, in comparison with the number of variables, and in order to respect the condition of stationarity of the EEG signal. This is, in fact, a necessary assumption in order to proceed with the spectral analysis of the signal. Then, three criteria were applied to those EEG epochs [64,65]: (i) Threshold criterion (amplitudes exceeding ± 100 μV); (ii) trend criterion (slope higher than 10 μV/s); (iii) sample-to-sample criterion (sample-to-sample amplitude difference higher than 25 μV). All EEG epochs marked as “artifact” were removed in order to have a clean EEG signal. In order to accurately define EEG bands of interest, for each participant, the individual alpha frequency (IAF) was computed on a closed eyes segment recorded prior to the experimental task. EEG recordings were segmented into trials, corresponding to the listening of each musical excerpt (ranging from approximately 8000 ms to 28,000 ms), and excluding all experimental phases, beyond the listening to music, potentially affected by muscular artifacts or cognitive processes for instance linked to the difficulty rating phase. The power spectrum density was calculated in correspondence of the different conditions with a frequency resolution of 0.5 Hz. Thus, the PSD was obtained for alpha (IAF − 2 ÷ IAF + 2 Hz) and gamma (IAF + 16 ÷ IAF + 25) [66]. Trials were normalized by subtracting the open eyes activity recorded before the beginning of the experimental task.

For the calculation of the average power for the right and left hemisphere, respectively, all the right (F4, F8, C4, Cp6, T8, P4, P8, O2) and all the left electrodes (F3, F7, C3, Cp5, T7, P3, P7, O1) were averaged. For the calculation of the power over the frontal area, F3, F7, F4, F8, and Fz electrodes were averaged. For the right frontal area power calculation, F4 and F8 electrodes were averaged, while for the left frontal area power calculation, F3 and F7 electrodes were averaged. Therefore, the frontal alpha asymmetry was calculated as the average power calculated over the right frontal area subtracted from the average power calculated over the left frontal area [45,67,68].

For the calculation of the lateralization index (LI), Formula (1), already employed in a previous study, was used [69]:(1)$$LI=\frac{R-L}{R+L}$$

The LI ranges from −1 to +1, with positive values implying more right relative asymmetry, while negative values more left relative asymmetry.

In case of the CI side, the formula was the following (2), already employed in a previous study [42]:(2)$$LI\;CI\;side\;based=\frac{CI\;side-NON\;CI\;side}{CI\;side+NON\;CI\;side}$$

### 2.5. Alexithymia

Alexithymia was assessed through the 20 items Toronto Alexithymia Scale (TAS-20) [50] validated for the Italian population [70] and already employed in the context of auditory stimuli both involving healthy participants [71] and tinnitus patients [72]. Each item is scored from 1 (strongly disagree) to 5 (strongly agree) for a maximum total of 100, and it includes three subscales: (a) difficulty in identifying feelings (DIF, difficulty in identifying feelings and distinguishing between emotional feelings and the bodily sensations of emotional arousal); (b) difficulty in describing feelings (DCF; difficulty finding words to express feelings to other); (c) EOT (externally oriented style). TAS-20 values were collected for each participant, and scoring was performed according to the published literature.

### 2.6. Statistical Analysis

For the statistical analysis, STATISTICA 12 software (StatSoft GmbH) was employed.

Significance level was defined as α = 0.05. The Shapiro–Wilk test was used in order to assess the normality of data distribution, that in a minority of cases did not fit the normal distribution, but it is worthy to note that, despite the statistical assumptions underlying the ANOVA methodology, it has been shown that ANOVA is not very sensitive to deviations from normality. In particular, simulation studies with non-normal distributions have shown that the false positive rate is not very affected by this violation of the normality assumption [73,74,75]. The UCI group was further divided into two groups: pre- and perilingual deaf CI users and postlingual deaf CI users, in order to compare these two groups for TAS-20 scores, age, and gamma power in the CI side and non-CI side hemisphere.

ANOVA tests were performed in the comparisons between groups for the different EEG indices (for all groups: right and left average gamma power for each hemisphere, gamma LI calculated concerning each hemisphere and the right and left frontal area, frontal alpha asymmetry and average right and left frontal alpha power; for UCI: CI-side and non-CI side average gamma power for each hemisphere, gamma LI calculated concerning each hemisphere and the CI-side and non-CI side average gamma power over the frontal areas); and for behavioural/declared data (correct responses, TAS-20). Three factors were investigated: group (two levels: UCI, NH), emotion (two levels: happy, sad), and SNR (three levels: quiet, SNR5, SNR10).

Unpaired *t*-tests were employed for the assessment of the differences between pre-/perilingual and postlingual groups for the variables: age, correct responses, TAS-20 scores, and average gamma power in the CI and non-CI side over the hemispheres and over the frontal areas. The unpaired *t*-test was also performed, comparing the left- and right-ear implanted patients in relation to the percentage of correct responses. Finally, the *t* test was used for comparing the two groups, NH and UCI, concerning the variable age.

Fisher’s exact test two-tailed was used for testing the eventual statistically significant differences between the frequency of the groups concerning nonalexithymic/alexithymic traits occurrence in the UCI and NH group, and within the UCI group (comparing pre-/perilingual and postlingual groups for the variables sex and nonalexithymic/alexithymic traits.

A logistic regression analysis between TAS-20 scores and sex for each group (UCI and NH) and subgroup (UCI: pre-/perilingual and postlingual) was performed in order to investigated the eventual relation between the mentioned continuous and dichotomic variables.

Pearson’s correlation analyses were performed between the EEG-based indices included in the study and some demographic, clinical, and behavioural variables for both UCI and NH groups (for the NH group, only the TAS-20 score, age, and percentage of correct responses were included in the analysis).

## 3. Results

### 3.1. Behavioural

The ANOVA test was conducted on the percentage of correct responses considering the factors: group (UCI/NH), emotion (sad/happy) and SNR (quiet/SNR10/SNR5). It resulted in a significant effect of the group (F(1, 26) = 9.306 *p* = 0.005; partial eta-squared = 0.263), with the NH group reporting a higher percentage of correct responses (96.35 ± 22.45%) in comparison to UCI (67.71 ± 22.45%). There was also an effect of emotion, with happy musical excerpts being recognized more than sad ones (F(1, 26) = 10.158, *p* = 0.004; partial eta-squared = 0.281). Furthermore, there was also a significant interaction between the variable group and emotion (F(1, 26) = 4.661, *p* = 0.040; partial eta-squared = 0.152), and in particular, the post hoc analysis showed that the percentage of correct responses reported by UCI for the sad musical excerpts was lower than the happy ones reported by the same group (*p* < 0.001), and it was also lower than the percentage of correct responses reported by the NH group for both happy and sad musical excerpts (*p* < 0.001 for both) (Figure 1). 

Within the UCI group, the *t*-test was performed comparing the left- and right-ear implanted patients in relation to the percentage of correct responses, resulting in no differences between them (t = −0.213 *p* = 0.833). Finally, we performed an analysis concerning the etiology of the deafness, collapsing percentages of correct responses data from pre- and perilingual deaf patients, and comparing them with the ones of postlingual deaf participants, resulting in no statistical differences between the two groups (t = −0.582 *p* = 0.568).

### 3.2. TAS-20

The comparison between NH and UCI groups did not show any difference concerning TAS-20 scoring (t = −0.166 *p* = 0.869) (Figure 2). Also, the assessment of the alexithymia occurrence in the two groups (NH: 1 alexithymic and 7 nonalexithymic/alexithymic traits; UCI: 6 alexithymic and 14 nonalexithymic/alexithymic traits) did not show statistical significance (Fisher’s exact test *p* = 0.633).

There was no difference in terms of alexithymia occurrence among CI users on the base of the etiology (pre-/perilingual and postlingual) of deafness (Table 1). In fact, Fisher’s exact test reported a lack of significance (*p* = 0.642) for the alexithymia frequency that occurred considering as first group the merging of pre- and perilingual deaf UCI participants, and as second group the postlingual deaf UCI patients (pre-/perilingual deafness: two alexithymic and seven nonalexithymic/alexithymic traits; postlingual: four alexithymic and seven nonalexithymic/alexithymic traits). 

In order to dig into the eventual influence of the etiology of deafness on TAS-20, a comparison between the groups was performed, subdividing UCI participants into pre- and perilingual deaf patients and the postlingual deaf group. Results did not show any statistical significance (TAS-20: t = 0.390 *p* = 0.701).

Concerning the side of the CI, an unpaired *t*-test was performed between right- and left-implanted CI users, and results did not show any statistical difference for the TAS-20 score (t = 0.717 *p* = 0.482). 

For the UCI group, we performed a correlation analysis between the TAS-20 score and percentages of correct responses and clinical variables, without evidencing statistical significances, specifically for the time of CI use (TAS-20: r = 0.130 *p* = 0.583), the residual hearing in the contralateral ear (TAS-20: r = 0.098 *p* = 0.680), the duration of deafness (TAS-20: r = 0.052 *p* = 0.826), and the percentage of correct responses (TAS-20: r = −0.142 *p* = 0.549).

Finally, for both UCI and NH groups, we performed an investigation of the correlation between TAS-20 score and age [76,77], returning a correlation only for the NH group, that is, higher TAS-20 scores were correlated with older ages (r = 0.862 *p* = 0.006), but not for the UCI one (r = 0.127 *p* = 0.593). It is important to highlight that there was not any statistically significant difference between the age of the participants belonging to the two groups (unpaired *t*-test t = −1.357 *p* = 0.186).

In order to investigate the eventual relation between TAS-20 scores and sex of the participants belonging to the two groups, a logistic regression analysis between TAS-20 scores and sex for each group was performed, resulting in any statistical significance for NH (chi-square = 0.281 df = 1 *p* = 0.596), and UCI (chi-square = 2.053 df = 1 *p* = 0.152). Moreover, the same analysis was also performed on the UCI subgroups, not resulting in any statistical significance, as well both for the pre-/perilingual group (chi-square = 1.189 df = 1 *p* = 0.275) and the postlingual group (chi-square = 0.714 df = 1 *p* = 0.398).

### 3.3. EEG

#### 3.3.1. Gamma Asymmetry

The ANOVA test was conducted on the LI gamma asymmetry values considering the factors group (UCI/NH), emotion (sad/happy), and SNR (quiet/SNR10/SNR5). The results did not show any statistical significance of the factors or interactions among them (all *p* > 0.05). Performing the same statistical analysis (ANOVA test considering the factors group, emotion, and SNR) separately on the left and right hemispheres gamma activity showed an effect of the factor emotion, with statistically significant higher gamma activity in correspondence of the listening to the happy musical excerpts than sad ones (left hemisphere: F(1, 26) = 5.819 *p* = 0.023, partial eta-squared = 0.183; right hemisphere: F(1, 26) = 8.851 *p* = 0.006, partial eta-squared = 0.254). Given the evidence of a lack of effect of the SNR, we performed on the UCI group an analysis of the gamma activity on the gamma asymmetry and on the right and left hemispheres separately, considering the factors emotion (sad/happy) and CI side (R/L). Results for the gamma asymmetry reported statistically significant right hemisphere lateralization for right-ear implanted patients, and left-lateralization for left-ear implanted participants (F(1, 18) = 33.789, *p* < 0.001; partial eta-squared = 0.652). Furthermore, when considering the two hemispheres separately, for both, there was a statistically significant effect of the factor emotion, with happy musical excerpts eliciting higher gamma activity than sad ones (left hemisphere: F(1, 18) = 7.735 *p* = 0.012; partial eta-squared = 0.300; right hemisphere: F(1, 18) = 6.359 *p* = 0.021; partial eta-squared = 0.261). Additionally, for only the gamma activity in the right hemisphere was there a statistically significant effect of the side of the CI, with higher PSD values for the left-ear implanted patients than for the right-ear implanted ones (F(1, 19) = 7.450 *p* = 0.014; partial eta-squared = 0.293).

In order to investigate whether there was a difference among the mean gamma activity comparing the hemispheres on the base of the traditional right/left lateralization and the CI/non-CI side, we performed an ANOVA considering the factor side that did not return any statistical significance (F(1, 19) = 1.981 *p* = 0.127; partial eta-squared = 0.094) (Figure 3).

In the UCI group, considering the side of the CI instead of the right/left categorization, an index of lateralization was calculated so as to assess the asymmetry between the CI and non-CI side, as defined by Formula (2) in the Methods section. On this index, we performed an ANOVA test, considering the factors emotion (sad/happy) and SNR (quiet/SNR10/SNR5). The results did not show any statistical significance of the factors or interactions among them (all *p* > 0.05). Furthermore, we calculated the mean of the gamma activity in the same hemisphere of the CI side and in the contralateral hemisphere. The ANOVA test considering the factors emotion (sad/happy) and SNR (quiet/SNR10/SNR5), both for the CI and non-CI side, showed statistically significant higher values for the happy compared to the sad music excerpts listening (respectively, F(1, 20) = 7.097 *p* = 0.015, partial eta-squared = 0.272; F(1, 19) = 17.051 *p* < 0.001, partial eta-squared = 0.473).

Moreover, for each group, we performed a correlation analysis between the mean gamma activity in the hemispheres (right/left and CI/non-CI side) and some demographic, clinical, and behavioural variables (for the NH group only the TAS-20 score, age, and percentage of correct responses were included in the analysis). In particular, a statistically significant negative correlation was found between gamma activity in the non-CI side and age at recording (r = −0.523 *p* = 0.018), and the TAS-20 score (r = −0.456 *p* = 0.043) (Figure 4), but not with the percentage of correct responses (r = −0.160 *p* = 0.499), with the residual hearing in the contralateral ear (r = −0.146 *p* = 0.539), with the time of CI use in months (r = −0.087 *p* = 0.714), or with the duration of deafness (r = 0.138 *p* = 0.562). Moreover, in the UCI group, a statistically significant negative correlation between the mean gamma activity in the right hemisphere and the age of the participants (r = −0.612 *p* = 0.004) was found. The abovementioned statistically significant correlation was, instead, not found in the UCI group when considering the gamma activity in the CI side, nor in the left and right sides for any of the just mentioned variables. Similarly, they were also not found in the NH group when performing the same analysis.

Finally, a similar analysis was also performed on the LI gamma, calculated considering both the right/left and UCI/non-UCI side and the abovementioned clinical and behavioural variables; the results showed a positive correlation between LI based on the CI side and the percentage of correct responses, that is, a correlation between a higher relative gamma activity on the CI side and the performance in the emotional recognition task (Pearson’s r = 0.448 *p* = 0.048) (Figure 5a). In addition, given the higher percentage of correct responses found for the happy in comparison to the sad musical excerpts, and the post hoc analysis on the interaction between the type of emotional content and the group showing such evidence holding true for the UCI group, we performed in all groups a correlation specifically between gamma LI and percentage of correct responses for happy and sad musical pieces separately. The results showed that only in the UCI group was there a positive correlation between gamma LI calculated, considering the side of the CI and the percentage of correct responses for the happy musical pieces (Pearson’s r = 0.521 *p* = 0.019) (Figure 5b).

Given the evidence in the UCI group of a negative correlation between the mean gamma power in the non-CI side and some of the abovementioned behavioural variables, in order to investigate such evidence in depth, within the UCI group we performed a comparison between pre- and perilingual deaf patients and postlingual ones, returning higher gamma activity in the non-CI side in the pre-and perilingual group in comparison to the postlingual one (t = 2.152 *p* = 0.045) (Figure 6). It is interesting to note that there was not any statistically significant correlation between period of CI use and gamma activity in both the CI side and non-CI side for either group (respectively: pre-/perilingual: r = −0.258 *p* = 0.502 and r = −0.038 *p* = 0.922; postlingual: r = 0.121 *p* = 0.724 and r = −0.238 *p* = 0.481). Similarly, in children CI users, no correlation was found between gamma (parietal) activation in working memory and hearing age (whereas the correlation with demographic age was obtained) [78].

#### 3.3.2. Frontal Gamma Asymmetry

The ANOVA test was conducted on the frontal gamma asymmetry values considering the factors group (UCI/NH), emotion (sad/happy), and SNR (quiet/SNR10/SNR5). The results evidenced a statistically significant interaction between the variables SNR and group (F(2, 52) = 3.333 *p* = 0.043; partial eta-squared = 0.114), and the post hoc analysis showed higher frontal gamma right lateralization for the SNR10 in comparison to the SNR5 (*p* = 0.012), while there was not any difference for all the other pairwise comparisons within and between groups (*p* > 0.05). 

When investigating the gamma activity over the right and left frontal areas separately, we performed an ANOVA test considering the factors group (UCI/NH), emotion (sad/happy), and SNR (quiet/SNR10/SNR5). The analysis showed higher gamma PSD values in response to the happy musical excerpts in comparison to the sad ones (F(1, 26) = 11.939 *p* = 0.002; partial eta-squared = 0.315) only for the right frontal area, but not for the left one (F(1, 26) = 3.270 *p* = 0.082; partial eta-squared = 0.112). 

Analogously to the analysis performed on the mean gamma asymmetry calculated over each hemisphere, for the frontal gamma asymmetry for each group we also performed a correlation analysis between the mean gamma activity over the frontal areas (right/left and CI/non-CI side) and some demographic, clinical, and behavioural variables. For the NH group, only the TAS-20 score, age, and percentage of correct responses were included into the analysis. In particular, in the UCI group, we did not find any correlation between the gamma activity in the right frontal area and TAS-20 score, time of CI use, percentage of correct responses, residual hearing in the contralateral ear, and duration of deafness. No statistical significance was found for the same Pearson’s correlation analysis performed on the left frontal area. For the NH group, no statistical significance was found, only a strong tendency towards a negative correlation between gamma activity over the right frontal area and the percentage of correct responses (Pearson’s r = −0.701, *p* = 0.053). Concerning the correlation tests performed on the CI and non-CI side, a negative correlation was found between the mean gamma activity over the non-CI side frontal area and the percentage of correct responses (Pearson’s r = −0.446, *p* = 0.049) (Figure 7). Furthermore, given the higher percentage of correct responses for the happy, in comparison to the sad, musical excerpts, and the Duncan post hoc analysis on the interaction between the type of emotional content and the group showing that the UCI group reported lower correct responses for the categorization of sad musical excerpts than happy ones (Figure 1), we performed in all groups a correlation specifically between the percentage of correct responses for happy and sad musical excerpts separately and frontal gamma activity in each right/left and CI and non-CI side. We, of course, limited the analysis to the right/left side for the NH group. The results showed for the UCI group showed a negative correlation between mean gamma activity in the non-CI side frontal area and the percentage of correct responses for the happy musical excerpts (Pearson’s r = −0.469, *p* = 0.037), but not for the sad ones (Pearson’s r = −0.350, *p* = 0.130). Moreover, for the NH group, a negative correlation was found between mean gamma activity over the right frontal area and the percentage of correct responses for the happy musical excerpts (Pearson’s r = −0.783, *p* = 0.022), but not for the sad ones (Pearson’s r = −0.577, *p* = 0.134).

Additionally, a similar analysis was also performed on the LI frontal gamma, calculated considering both the right/left and CI/non-CI side, and the abovementioned clinical and behavioural variables. The results did not show statistical significances for the UCI or the NH groups (*p* > 0.05).

Finally, similarly to the analysis already performed on the two hemisphere sides in gamma activity, given the evidence in the UCI group of a negative correlation between the mean gamma activity over the frontal area of the non-CI side and some of the abovementioned behavioural variables, in order to dig into such a result, within the UCI group, we performed a comparison between pre- and perilingual deaf patients and postlingual ones, evidencing a not-statistically-significant higher gamma activity in the non-CI side in the pre-and perilingual group in comparison to the postlingual one (t = 0.906, *p* = 0.377).

#### 3.3.3. Frontal Alpha Asymmetry

The ANOVA test was conducted on the frontal alpha asymmetry values considering the factors group (UCI/NH), emotion (sad/happy), and SNR (quiet/SNR10/SNR5). The results showed more positive values in correspondence to listening to happy musical excerpts (F(1, 26) = 4.219, *p* = 0.050’ partial eta-squared = 0.140). Furthermore, there was a significant interaction between the variable group and emotion (F(1, 26) = 9.379, *p* = 0.005; partial eta-squared = 0.265), with the UCI group showing negative values for both happy and sad musical excerpts, while the NH group showed positive values for both, and, in particular, with a within-group increase in response to happy musical excerpts in comparison to sad ones (*p* = 0.001). Given the evidence of a lack of effect of the SNR, we performed on the UCI group an analysis of the frontal alpha asymmetry and of the alpha activity in the right and left hemisphere separately, considering the factors emotion (sad/happy) and CI side (R/L). The results did not show for the frontal alpha asymmetry any effect of the CI side (F(1, 18) = 0.049, *p* = 0.827; partial eta-squared = 0.003), while for both the left and right frontal areas, there was a statistically significant effect of the emotion, with happy musical excerpts eliciting higher alpha activity than sad ones (left hemisphere: F(1, 18) = 18.772, *p* < 0.001, partial eta-squared = 0.510; right hemisphere: F(1, 18) = 10.687, *p* = 0.004, partial eta-squared = 0.372).

Pearson’s correlation analyses were performed between the mean alpha activity in the frontal areas (right/left and UCI/non-UCI side) and some demographic, clinical, and behavioural variables for both UCI and NH groups (for the NH group, only the TAS-20 score, age, and percentage of correct responses were included into the analysis), without evidencing any statistical significance.

Finally, a similar analysis was performed also on the frontal alpha asymmetry, calculated considering both the right/left and CI/non-CI side, and the abovementioned clinical and behavioural variables. The results did not show statistical significances for the UCI or the NH group.

Also, for the frontal alpha asymmetry index and the average of the alpha activity over the right and left or CI and non-CI side, we performed a Pearson’s correlation analysis with behavioural and clinical data (these latter only in the case of the UCI group), without evidencing any statistical significance.

## 4. Discussion

The aim of the present study was the investigation, through a neuroscientific approach, of the neural correlates of emotion recognition skills apart from language in adult UCI users during a music in noise recognition task, and the investigation of the eventual comorbidity of deafness with alexithymia (as indicated by TAS-20 scores). Such aims were achieved, in fact; despite the results showing no effect of the background noise, they supported that EEG gamma activity related to emotion processing in the UCI group presented alterations in comparison to the NH group, and that such alterations were also modulated by the deafness etiology. Indeed, the main results showed that the UCI group obtained poorer performances in emotion recognition than the NH controls, and there was a negative correlation between TAS-20 scores and mean EEG gamma activity reported in the hemisphere contralateral to the CI side. Moreover, there was a correlation between the LI based on the CI side gamma activity values over the hemisphere and the percentage of correct responses. Finally there was a correlation between the average gamma activity estimated in the frontal area contralateral to the CI side and the percentage of correct responses in the UCI group.

The finding of a deficit in emotion recognition in CI users in comparison to NH controls is robust evidence, supported by many papers both in adult and children [19,26,79,80,81] populations.

The evidence of better performance in the recognition of the happy content in comparison to the sad one of musical stimuli (Figure 1) has been already shown [56] and suggested to be linked to higher levels of alpha activity in comparison to sad excerpts in the parietal area and F7 and F8 EEG channels [82]. In the present study, such evidence was extended to gamma band activity, showing a negative correlation between the percentage of correct responses when categorizing happy musical excerpts and the levels of gamma activity in the right frontal area for NH participants, while for the UCI group, there was a negative correlation between such percentages of correct responses for happy stimuli and both levels of gamma activity in the non-CI side frontal area, and LI gamma calculated on the basis of the CI side (Figure 5b). Concerning EEG gamma activity assessed in correspondence of the emotion recognition task, a previous study comparing two CI strategies for prosodic cues recognition showed that gamma activity would reflect top-down cognitive control during such a task and that the strategy linked to such gamma activity elicitation was also more performed in regard to happy prosody perception [41]. This evidence is really interesting in connection with the present study because of the correlation between the percentage of correct responses in emotion recognition, in general, for all the stimuli and, in particular, for the happy stimuli and relative higher gamma activity (indicated by the LI) in the non-CI side of CI users. Moreover, it is interesting because of the negative correlation between frontal gamma activity in the non-CI side and the percentage of correct responses for happy stimuli for the UCI group, while in the NH group, a negative correlation was shown between right frontal gamma activity and the percentage of correct responses for happy stimuli.

Concerning hemispheric gamma activity in relation to emotion recognition, in the present study, right hemisphere lateralization was found for right-ear implanted patients, and left-lateralization for left-ear implanted participants (*p* < 0.001). Interestingly, employing the LI on UCI children, it has been suggested an alteration of such asymmetry in child CI users in comparison to NH controls, in particular in relation to higher right hemisphere gamma activity, associated with lower emotion recognition abilities [42]. The important difference between the category of stimuli employed in the present study and the one employed in the just-mentioned research [42] is that musical stimuli are used here, whereas human nonverbal vocalizations (e.g., laughter, surprise) were used there. This could be at the basis of the different results, given the different evolutionary meaning of these two databases, possibly in conjunction with the different age and composition of the two groups in terms of etiology (in the present study, participants recruited were pre-, peri-. and postlingual deaf patients, whilst in [42], participants were only pre- and perilingual deaf patients). In fact, the present research also showed a negative correlation between the non-CI side gamma activity and the age, but not the period of deafness or the period of CI use. In fact, in [42], the side of the CI did not influence the investigated EEG rhythms, instead showing only an effect of the hemisphere, explained by the higher right hemisphere specialization for emotion processing [83]. Conversely, in the present study, the higher lateralization is, instead, influenced by the side of the CI. With respect to this, the LI based on the CI side showed a positive correlation with the percentage of correct responses (Figure 5), that is, the higher the relative asymmetry in gamma activity toward the CI side, the higher the percentage of correct responses. Conversely, in [42], the percentage of correct responses was instead correlated to LI gamma based on the right–left asymmetry (relative higher right gamma activity correlated to higher percentages of correct responses), independently of the CI side. Moreover, in the present adult UCI sample, gamma activity in the non-CI side was negatively correlated to TAS-20 scores (Figure 4) and with age. Furthermore, such gamma activity in the non-CI side was higher in the pre-/perilingual UCI group in comparison to the postlingual group (Figure 6). The sum of such results would support that gamma activity related to emotion processing in the adult UCI group presents alterations in comparison to the NH group, and that such alterations are also modulated by the deafness etiology. Furthermore, given the lack of correlation between the non-CI side gamma activity with the time of CI use, the results strongly support that the increased gamma activity in the non-CI side of the pre-/perilingual group in comparison to the postlingual group could be explained by different compensatory plasticity mechanisms between congenital and acquired hearing loss, in particular, not arising from a time-dependent CI-induced plasticity, but to developmental factors. In fact, despite that a certain degree of plasticity has been suggested also in adulthood after cochlear implantation [84,85,86,87,88], congenital sensory deprivation produces massive alterations in brain structure, function, connection, and neural interaction [89]. Moreover, many forms of plasticity are adaptive and instrumental to optimize performance [86], and this is suggested in the present study by the negative correlation between gamma activity in the non-CI side and the TAS-20 scores, implying higher gamma activity in the non-CI side linked to lower alexithymia impairments. Therefore, the already mentioned higher gamma activity in the non-CI side of the pre-/perilingual group in comparison to the postlingual group could be a form of adaptive plasticity attempting to cope with the potential deficits in emotion processing due to auditory deprivation during early development. In fact, for instance, the integration between auditory and visual cues is fundamental for the further acquisition of emotional processing skills [1]. In particular, relatively higher gamma activity in the CI side corresponds to positive processes, correlated to higher capacities in emotion recognition, while the gamma activity in the contralateral side (non-CI side) is possibly linked to positive processes too, being inversely correlated to alexithymia symptoms (as indexed by the TAS-20 score), and it is also inversely correlated to age; in fact, in the elderly, alexithymia is associated with age in the general population [76,77], as also found in the present NH sample (correlation between TAS-20 and age), but not in the UCI group. 

Emotional prosody, one of the most studied topics in CI users concerning emotion recognition, is the capacity to express emotions by variations of pitch, intensity, and duration [90]. Given the wide research concerning prosody, for the discussion of the present results, we will also refer to such studies, although in the present protocol, musical emotional stimuli were employed. However, this employment was supported by the demonstrated analogy between music and language [91,92,93]. With respect to prosody, brain activity in the gamma-band range should indicate facilitated prosody recognition modulated by strategy; in particular, a peak of gamma band activity (40 Hz) was found mostly in the anterior temporal left area for correctly recognized words [94]. This could be reflected in our study by the negative correlation between right frontal gamma activity and percentage of correct responses for happy musical stimuli (and just missing the statistical significance for the general percentage of correct responses) in NH participants. In fact, as already shown in a previous study [82], given the use of musical emotional stimuli in the study, instead of words, as in [94], the analogue brain processes seem to be localized in the right instead of the left hemisphere. Moreover, in a study involving CI users during a verbal working memory task, only in the NH control group was gamma activity found to be localized in the parietal area, supporting auditory verbal working memory [78]. Furthermore, an fNIRS study showed that recognition of vocal emotional stimuli appeared in the right supramarginal gyrus (belonging to the parietal lobe) after CI implantation in infants and that for the development of emotion processing capabilities, different timeline and neural processing occur from those in NH peers [80]. In addition, the right supramarginal gyrus appears to be involved in empathy [95], and it appears to be modulated by age [96], analogously to the here-retrieved gamma activity in the non-CI side. In the present study, 9 participants were pre-/perilingual deaf patients, analogous to the infants and toddlers included in the abovementioned article, while 11 were postlingual deaf patients. Interestingly, a statistically significant higher gamma activity was found in the pre-/perilingual deaf group in the hemisphere contralateral to the CI side, which was negatively correlated with TAS-20 score and age in the general UCI sample, suggesting the occurrence of “facilitating EEG” activity in the processing of emotional stimuli. This could possibly be due to longer and wider compensatory plasticity mechanisms’ occurrence compared to the postlingual deaf group, in order to achieve similar performance in emotion recognition skills [97].

Concerning the frontal alpha asymmetry results, there was a higher approach tendency (as indicated by the positive values of the index [67]) toward happy musical stimuli than to sad ones, and, interestingly, happy stimuli were also more recognized than the sad ones by the present sample. Moreover, the NH group showed higher approach tendency to happy musical stimuli than to sad ones, whilst the UCI group did not report statistically significant differences, supporting a lack of sensitivity in the recognition of music in comparison to NH participants, already evidenced in a previous study employing frontal alpha asymmetry in UCI adults watching musical videos [45].

Finally, in general, there was no effect of the background noise on the results, differently from the previous paper [82], because in that case, the effect of the emotional content had been ideally removed because of the subtraction of the baseline from the different SNRs. Additionally, in that study, the listening effort was the object of investigation, estimated through parietal and frontal (F8) alpha indices, and not gamma, as in the present study. However, the results apparently conflict with a previous study concerning emotional prosody recognition in a vocoded speech, reporting a decrease in recognition performance with worsening SNR [98]. Moreover, it is important to highlight that Morgan and colleagues reported that for influencing emotion recognition performances, less favorable SNRs were required (−15, −10, −5 dB SNR) than the ones employed in the present study (SNR5 and 10), but it must be underlined that such SNRs are poorly feasible in CI users; in fact, Morgan and coworkers tested NH participants. Finally, it must be considered that the type of the employed emotional stimuli was different: musical excerpts in the present case and verbal prosodic stimuli in the just-mentioned research. 

As future developments, in order to perform a thorough evaluation of the CI users and of the deaf condition in general, it would be interesting to assess, through neuroscientific methods, the eventual influence of the emotional processing alterations found in the present study in the UCI group and also on other sensory modalities known to be particularly linked to emotional experience, such as olfaction [99,100] and taste [101,102].

The present study, as with every research, of course presents limitations, like, for instance, the limited number of participants and the analysis concerning pre-/perilingual and postlingual deaf patients that would benefit from an enlargement of the sample, given the limited sample size tested. However, given the balanced number of CI users when considering pre-/perilingual etiology (*n* = 9) and postlingual etiology (*n* = 11), it could be argued that the eventual influence of such a factor would be mitigated in the present study. Future studies could explore these differences further. Moreover, the eventual influence of the bilateral CI condition in the evaluated phenomena should be investigated, in particular concerning the gamma activity patterns during emotion recognition tasks, given the already suggested differences between unilateral and bilateral CI users [57]. In fact, we could expect a greater similarity concerning EEG patterns between bilateral CI users and NH controls than between unilateral CI users and the NH group, as already suggested, for instance, for frontal alpha asymmetry patterns [45] and for theta and gamma neural correlates of working memory [103]. Finally, studies concerning emotion recognition and production in CI users appear particularly worthy for patients’ wellbeing, and to be taken into account for practical applications like rehabilitation programs, as very recently underlined [104].

## 5. Conclusions

EEG gamma activity appears to be fundamental to the processing of the emotional aspect of music and also to the psychocognitive emotion-related component in adults with CI. Although there was not a statistically significant difference in alexithymia scores between the UCI and NH groups, but emotion recognition performance was higher in NH compared to UCI participants, the neural correlates in the gamma band suggest the occurrence of compensatory neuroplasticity mechanisms trying to counteract sensory-deprivation-induced deficits in emotion processing in the UCI group. In particular, relative higher gamma activity in the CI side corresponds to positive processes correlated with higher emotion recognition abilities, whereas gamma activity in the non-CI side may be related to positive processes inversely correlated with alexithymia and also inversely correlated with age. Therefore, gamma patterns seem to be a neurophysiological signature that accompanies the life of the CI patient from childhood to adulthood, apparently modifying itself, as suggested by the different results obtained in different studies.

## Figures and Tables

**Figure 1 brainsci-14-00927-f001:**
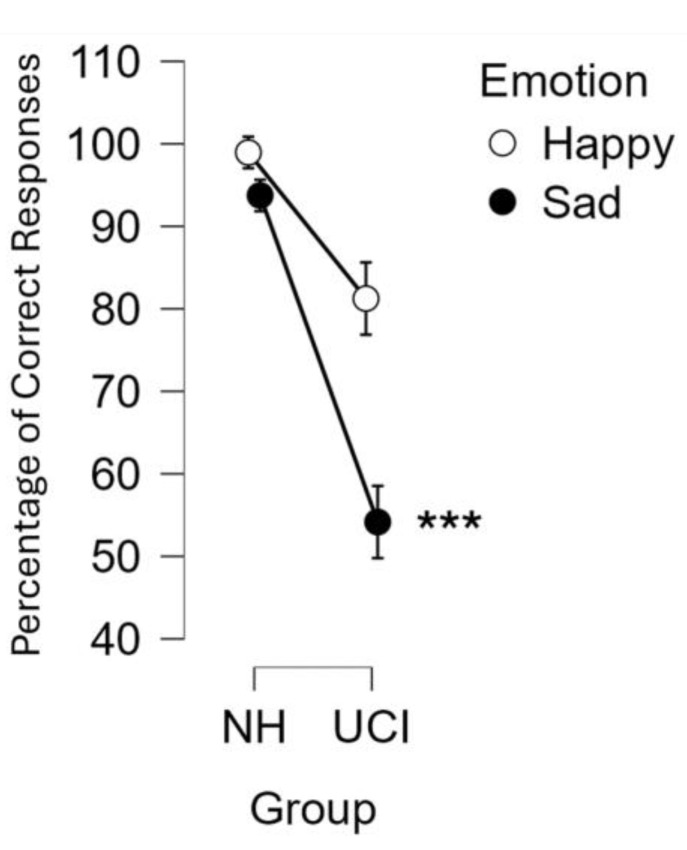
The graph represents the percentage of correct responses in the categorization of happy and sad musical excerpts for each group: unilateral cochlear implant users (UCI) and normal hearing (NH) controls. The interaction between the variables emotion (happy/sad) and group (UCI/NH) was statistically significant (*p* = 0.040). Error bars stand for standard error. *** stands for significance level of the post hoc comparisons of *p* ≤ 0.001.

**Figure 2 brainsci-14-00927-f002:**
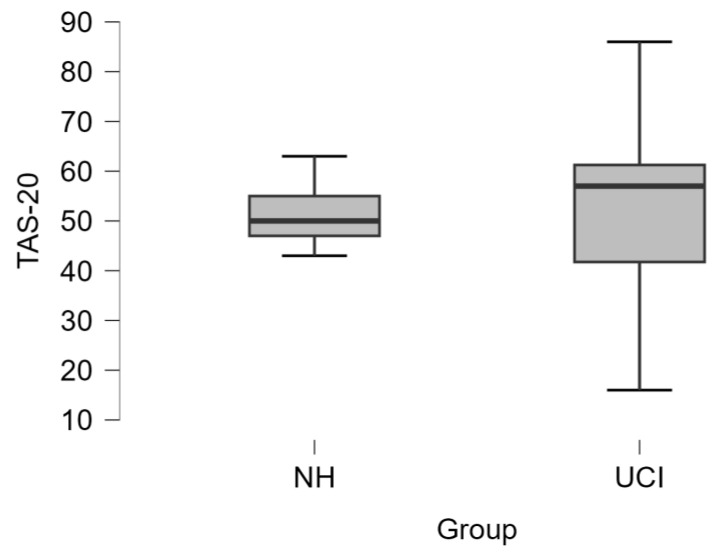
The boxplot represents the comparison between NH and UCI group for the Toronto Alexithymia Scale (TAS-20) scores, which did not report any statistically significant difference (t = −0.166, *p* = 0.869).

**Figure 3 brainsci-14-00927-f003:**
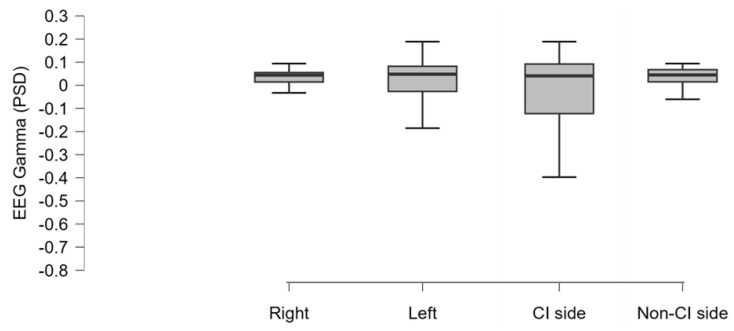
The boxplot represents the mean EEG gamma activity (PSD: power spectral density) estimated in the UCI group and resulting from the averaging of the signal acquired from the electrodes located in each hemisphere (right, left. CI side, non-CI side), as specified in the Methods section. No statistically significant differences were evidenced.

**Figure 4 brainsci-14-00927-f004:**
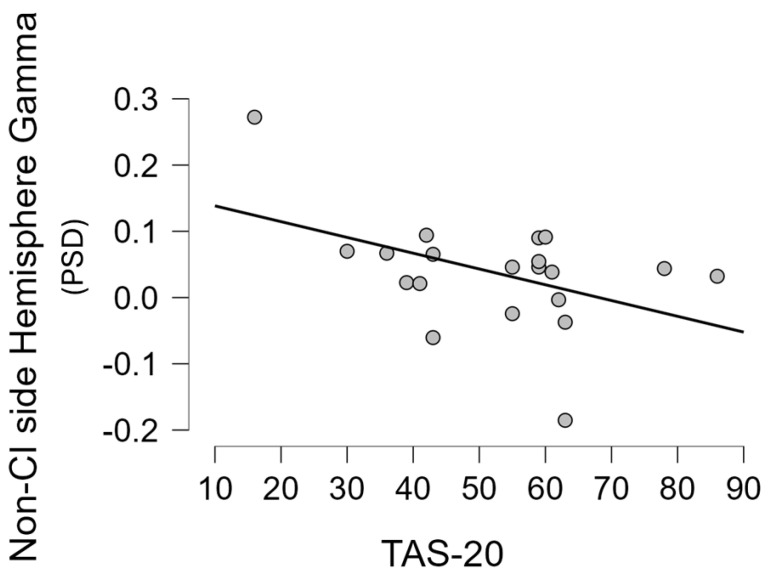
The scatterplot represents the correlation (Pearson’s r = −0.456 *p* = 0.043) between the mean EEG gamma activity reported in the hemisphere contralateral to the CI side and the alexithymia (Toronto Alexithymia Scale—TAS-20 scores) in the UCI group. PSD: power spectral density.

**Figure 5 brainsci-14-00927-f005:**
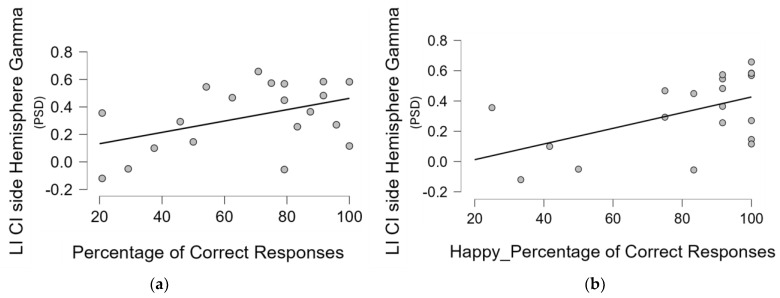
These scatterplots represent, for the UCI group, respectively, the correlation between the lateralization index (LI) based on the CI side gamma activity (PSD: power spectral density) values over the hemisphere and the percentage of correct responses irrespective of the emotional content of the musical excerpts (**a**), and the percentage of correct responses only for the happy musical excerpts (**b**).

**Figure 6 brainsci-14-00927-f006:**
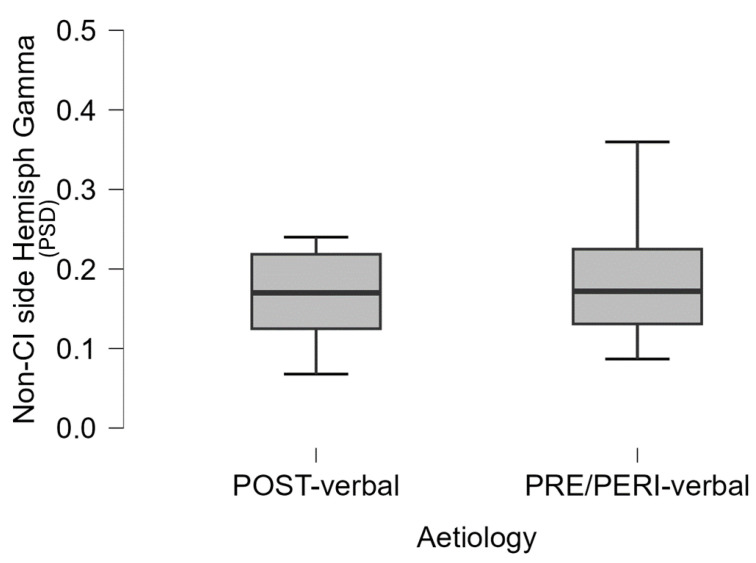
The boxplot represents the difference between postlingual and pre-/perilingual deaf UCI with respect to the mean gamma activity (PSD: power spectral density) in the non-CI side hemisphere (*p* = 0.045).

**Figure 7 brainsci-14-00927-f007:**
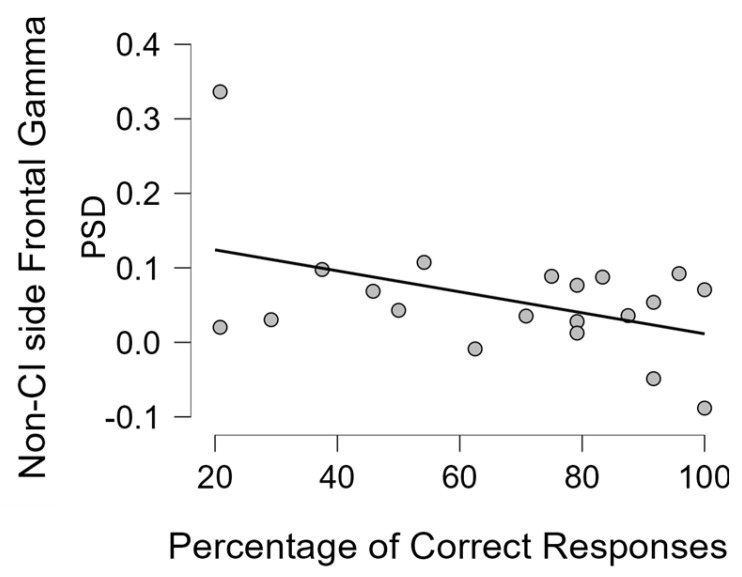
The scatterplot represents the statistically significant correlation (Pearson’s r = −0.446, *p* = 0.049) between the average gamma activity (PSD: power spectral density) estimated in the frontal area contralateral to the CI side and the percentage of correct responses in the UCI group.

**Table 1 brainsci-14-00927-t001:** The table reports the frequencies of alexithymia in the pre-/perilingual deaf UCI participants and in the postlingual deaf UCI participants. Fisher’s exact test did not show any statistical significance concerning such distribution (*p* = 0.642).

UCI GROUP	ALEXITHYMIC	NON-ALEXITHYMIC/ALEXITHYMIC TRAITS
PRE-/PERI-LINGUAL	2 (22.22%)	7 (77.78%)
POST-LINGUAL	4 (36.36%)	7 (63.64%)

## Data Availability

The data presented in this study are available on request from the corresponding author. The data are not publicly available due to privacy reasons.

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
