# Peer review of "Investigation of Deficits in Auditory Emotional Content Recognition by Adult Cochlear Implant Users through the Study of Electroencephalographic Gamma and Alpha Asymmetry and Alexithymia Assessment"

_brainsci, 2024, doi:10.3390/brainsci14090927_

Round 1
Reviewer 1 Report
Comments and Suggestions for Authors
Summary: This is an interesting and important study which examined how EEG gamma power is related to emotional music processing in unilateral cochlear implant users in comparison to normal hearing adults. Lateralization was examined in CI users by examining hemispheric differences as well as differences based on side of implantation. Additionally, the authors examined differences based on age of deafness onset.
Major Comments:
1. The authors report the effects of age in the results section. While the ages of the UCI and NH groups are mentioned in the methods section, the authors should indicate whether there were significant age differences between the groups.
2. More details about the UCI group are needed. Did any participants use hearing aids in the contralateral ear? What was the degree of hearing loss in the contralateral ear? At what ages did they obtain CIs? What was the duration of deafness before CI use? Did they use hearing aids prior to CI use?
3. The authors examine correlations with the duration of deafness, but this should be properly defined. Was the duration of deafness defined as the time between the onset of hearing loss and obtaining any amplification, or only before obtaining the CI?
4. The authors later separated the UCI group into pre/peri-lingual deafness and post-lingual deafness groups. It would be helpful to provide the ages and gender composition of these subgroups and indicate whether there were significant age differences between them.
5. In the discussion, the authors attribute differences in NON-CI side gamma activity between the pre/peri-lingual and post-lingual deafness groups to longer CI use. However, there were no correlations reported between gamma activity and the duration of CI use. The authors should address this in their discussion and consider the potential differences in compensatory plasticity between congenital and acquired hearing loss. It seems plausible that the effects of hearing loss may differ depending on when in development the hearing loss occurred.
6. The discussion could be strengthened by including additional avenues for future research and acknowledging the limitations of the study. For example, the analysis of pre- vs post-lingually deafened individuals is limited by the sample size (n=9 and n=11, respectively). Future research could explore these differences further.
Minor Comments:
1. The English in this paper is difficult to understand, with numerous grammatical errors and typos. Many sentences are overly long, making them challenging to follow.
a. Lines 49-52: Consider breaking this into two sentences.
b. Fix this sentence: “It is obvious the not physiological development and functioning of auditory system in congenitally deaf children who restored their hearing through cochlear implant” (lines 54-55).
c. When discussing the aim of the study throughout the paper the word the is missing before aim
d. I suggest breaking lines 185-188 into two sentences.
e. Throughout the results section the authors use “it was found” incorrectly.
f. Change has been showed or have showed to has been shown or have shown throughout the manuscript.
g. Consider breaking lines 573-581 into multiple sentences.
2. The authors note a strong tendency for a negative correlation between right frontal gamma activity and correct responses but do not report the p-value. Including this information would aid in evaluating the significance of this statement.
3. In the introduction, the authors mention differences in emotional processing between unilateral and bilateral CI users. It could be valuable to discuss how the results of this study might compare to those of bilateral CI users in the discussion section.
4. The conclusions could be more impactful by highlighting a main finding, rather than just stating that gamma activity is related to emotional music processing in adults with CIs.
5. For Figure 1. It is unclear if “Emotion” is the title of the figure or the key. Consider moving it over the key or centering it over the graph for clarity.
6. Line 490 fix {Citation}
Comments on the Quality of English Language
1. The English in this paper is difficult to understand, with numerous grammatical errors and typos. Many sentences are overly long, making them challenging to follow.
a. Lines 49-52: Consider breaking this into two sentences.
b. Fix this sentence: “It is obvious the not physiological development and functioning of auditory system in congenitally deaf children who restored their hearing through cochlear implant” (lines 54-55).
c. When discussing the aim of the study throughout the paper the word the is missing before aim
d. I suggest breaking lines 185-188 into two sentences.
e. Throughout the results section the authors use “it was found” incorrectly.
f. Change has been showed or have showed to has been shown or have shown throughout the manuscript.
g. Consider breaking lines 573-581 into multiple sentences.
Author Response
REVIEWER 1
Summary: This is an interesting and important study which examined how EEG gamma power is related to emotional music processing in unilateral cochlear implant users in comparison to normal hearing adults. Lateralization was examined in CI users by examining hemispheric differences as well as differences based on side of implantation. Additionally, the authors examined differences based on age of deafness onset.
Major Comments:
- The authors report the effects of age in the results section. While the ages of the UCI and NH groups are mentioned in the methods section, the authors should indicate whether there were significant age differences between the groups.
We thank the Reviewer for the comment and apologise for the lack of clarity: there wasn’t any statistically significant difference between the UCI and NH group, in fact an unpaired t-test performed on the age of the two groups resulted in t= -1.357 p= 0.186. We added such detail in the manuscript in the results section as follows at the end of the 3.2 section:
“It is important to highlight that there wasn’t any statistically significant difference between the age of the participants belonging to the two groups (unpaired t-test t= -1.357 p= 0.186).”
- More details about the UCI group are needed. Did any participants use hearing aids in the contralateral ear? What was the degree of hearing loss in the contralateral ear? At what ages did they obtain CIs? What was the duration of deafness before CI use? Did they use hearing aids prior to CI use?
We thank the Reviewer for the comment and apologise for the lack of clarity: none of the participants wore hearing aids in the contralateral ear during the test. We specified such detail in the Sample paragraph within the Methods section as follows:
“During the test, none of the CI users wore hearing aids in their contralateral ear.”
In addition, concerning the degree of hearing loss in the contralateral ear to the implanted one presented an average PTA (calculated including the frequencies 250-500-1000-2000-4000 Hz) ± standard deviation: 101.05 ± 13.77 dB. We added such detail in the Methods section in the Sample paragraph as follows:
“Concerning the UCI group, the degree of hearing loss in the contralateral ear to the implanted one presented an average PTA ± standard deviation: 101.05 ± 13.77 dB.”
Concerning the age at implantation, the duration of deafness before CI use and if they employed hearing aids before CI use, we added such details in the Sample paragraph in the Methods section as follows:
“Furthermore, in the UCI group, the age at implantation was 40.40 ± 14.59 years old, the duration of deafness (from the onset of deafness to implantation date) was 25.25 ± 14.64 years, the period of CI use at the time of the experiment was 5.16 ± 6.17 years, and all patients used hearing aids prior to CI use, according to Italian guidelines for cochlear implantation (https://www.iss.it/-/impianto-cocleare-adulto-bambino).”
- The authors examine correlations with the duration of deafness, but this should be properly defined. Was the duration of deafness defined as the time between the onset of hearing loss and obtaining any amplification, or only before obtaining the CI?
We thank the Reviewer for the comment and apologise for the lack of clarity. The comment in fact allowed us to specify such detail in the Methods section, as reported in the response to the previous point. In the present study, according to the definition shared by the Clinicians involved in the present study, performed in different Clinics in Italy, the duration of deafness was defined as the period occurring from the onset of deafness to the implantation date.
- The authors later separated the UCI group into pre/peri-lingual deafness and post-lingual deafness groups. It would be helpful to provide the ages and gender composition of these subgroups and indicate whether there were significant age differences between them.
We thank the Reviewer for the comment and apologise for the lack of clarity. We added such information in the Sample paragraph in the Methods section, as follows:
“The UCI group was also divided into two subgroups on the base of the deafness etiology: pre/peri-lingual (6F, 3M; mean age ± st. dev.: 37.44 ± 12.85) and post-lingual group (5F, 6M; mean age ± st. dev.: 52.82 ± 7.68), of course presenting a statistically significant increase concerning age for the post-lingual group in comparison to the pre-lingual one (t=-3.321 p=0.004), justified by the fact that typically in Italy is more frequent to have younger pre-/peri-lingual adult CI users thanks to the quite recent implementation and improvement over all the Italian territory of the national neonatal screening program that allows access to early auditory rehabilitation to younger patients [1]. In accordance to this, and with the update of the guidelines for the definition of CI candidates, it is in fact expected a progressive decrease in the proportion of candidates for a CI with long durations of profound deafness, because patients undergo CI surgery earlier in the time course of their deafness [2]”
- In the discussion, the authors attribute differences in NON-CI side gamma activity between the pre/peri-lingual and post-lingual deafness groups to longer CI use. However, there were no correlations reported between gamma activity and the duration of CI use. The authors should address this in their discussion and consider the potential differences in compensatory plasticity between congenital and acquired hearing loss. It seems plausible that the effects of hearing loss may differ depending on when in development the hearing loss occurred.
We thank the Reviewer for the valuable comment and apologise for the lack of clarity. According we added results concerning the correlation between period of CI use and gamma activity in both the CI and non-CI side, at the end of the 3.3.1 paragraph as follows:
“It is interesting to note that there wasn’t any statistically significant correlation between period of CI use and gamma activity in both the CI side and NON-CI side for neither group (respectively: pre-/peri-lingual: r=-0.258 p=0.502 and r=-0.038 p=0.922; post-lingual: r=0.121 p=0.724 and r=-0.238 p=0.481). Similarly, in children CI users no correlation was found between gamma (parietal) activation in working memory and hearing age (whereas the correlation with demographic age was obtained) [3].”
Given such lack of correlation with the time of CI use strongly supports, as suggested by the Reviewer, that the increased gamma activity in the NON-CI side of the pre-/peri-lingual group in comparison to the post-lingual group could be explained by different compensatory plasticity mechanisms between congenital and acquired hearing loss, in particular, not arising from a time dependent CI-induced plasticity, but to developmental factors. In fact, despite a certain degree of plasticity has been suggested also in adulthood after cochlear implantation [4], [5], [6], [7], [8], congenital sensory deprivation produces massive alterations in brain structure, function, connection and neural interaction [9]. Moreover, many forms of plasticity are adaptive, instrumental to optimize performance [6], and this is suggested in the present study by the negative correlation between gamma activity in the NON-CI side and the TAS-20 scores, implying higher gamma activity in the NON-CI side linked to lower alexithymia impairments. Therefore, the already mentioned higher gamma activity in the NON-CI side of the pre-/peri-lingual group in comparison to the post-lingual group could be a form of adaptive plasticity attempting to cope the potential deficits in emotion processing due to auditory deprivation during early development, in fact for instance the integration between auditory and visual cues is fundamental for further acquisition of emotional processing skills [10]. We addressed the Reviewer’s point adding the following period in the Discussion section:
“Moreover, given the lack of correlation between NON-CI side gamma activity with the time of CI use, results strongly supports, that the increased gamma activity in the NON-CI side of the pre-/peri-lingual group in comparison to the post-lingual group could be explained by different compensatory plasticity mechanisms between congenital and acquired hearing loss, in particular, not arising from a time dependent CI-induced plasticity, but to developmental factors. In fact, despite a certain degree of plasticity has been suggested also in adulthood after cochlear implantation [4], [5], [6], [7], [8], congenital sensory deprivation produces massive alterations in brain structure, function, connection and neural interaction [9]. Moreover, many forms of plasticity are adaptive, instrumental to optimize performance [6], and this is suggested in the present study by the negative correlation between gamma activity in the NON-CI side and the TAS-20 scores, implying higher gamma activity in the NON-CI side linked to lower alexithymia impairments. Therefore, the already mentioned higher gamma activity in the NON-CI side of the pre-/peri-lingual group in comparison to the post-lingual group could be a form of adaptive plasticity attempting to cope the potential deficits in emotion processing due to auditory deprivation during early development, in fact for instance the integration between auditory and visual cues is fundamental for further acquisition of emotional processing skills [10]”
- The discussion could be strengthened by including additional avenues for future research and acknowledging the limitations of the study. For example, the analysis of pre- vs post-lingually deafened individuals is limited by the sample size (n=9 and n=11, respectively). Future research could explore these differences further.
We thank the Reviewer for the useful suggestion and added at the end of the Discussion section the following statements:
“The present study, as every research, of course present limitations, like for instance the limited number of participants and the analysis concerning pre-/peri-lingual and post-lingual deaf patients that would benefit from an enlargement of the sample, given the limited sample size tested. However, given the balanced number of CI users when considering pre-/peri-lingual etiology (n=9) and post-lingual etiology (n=11), it could be argued that the eventual influence of such factor would be mitigated in the present study. Future studies could explore these differences further. Moreover, it should be investigated the eventual influence of the bilateral CI condition in the evaluated phenomena, and in particular concerning the gamma activity patterns during emotion recognition tasks, given the already suggested differences between unilateral and bilateral CI users [11]. In fact, it could be expected a greater similarity concerning EEG patterns between bilateral CI users and NH controls than between unilateral CI users and NH group, as already suggested for instance for frontal alpha asymmetry patterns [12] and for theta and gamma neural correlates of working memory [13]. Finally, studies concerning emotion recognition and production in CI users appear particularly worthy for patients’ wellbeing, and to be taken into account for practical applications like rehabilitation programs, as very recently underlined [14].”
Minor Comments:
- The English in this paper is difficult to understand, with numerous grammatical errors and typos. Many sentences are overly long, making them challenging to follow.
We thank the Reviewer for the comment and apologise for the perceived poor quality of English. We worked on the text and we feel it has been improved. We hope it will satisfy the Reviewer.
- Lines 49-52: Consider breaking this into two sentences.
We thank the Reviewer for the suggestion and modified the sentence as follows:
“Emotions represent a key item in communication purposes. In fact, from infancy, the processing emotional expressions would develop into the capability of recognizing different emotions, by leveraging personal experience and through the maturation of sensory and perceptual systems, particularly auditory and visual ones.”
- Fix this sentence: “It is obvious the not physiological development and functioning of auditory system in congenitally deaf children who restored their hearing through cochlear implant”(lines 54-55).
We thank the Reviewer for the suggestion and modified the sentence as follows:
“It is obvious that the development and functioning of the auditory system in congenitally deaf children who have recovered their hearing through cochlear implants (CIs) is not physiological”
- When discussing the aim of the study throughout the paper the word theis missing before aim
We thank the Reviewer for the correction, we modified along the text accordingly.
- I suggest breaking lines 185-188 into two sentences.
We thank the Reviewer for the suggestion, we modified along the text accordingly as follows:
“Each list included 24 musical stimuli, composed by 8 musical excerpts, half happy and half sad, belonging to the original database, delivered in Quiet (Q), and at two signal to noise ratio (SNR) 10 and 5. The employed background noise for SNR conditions was continuous 4-talker babble background noise [15], with SNR5 being the most difficult audibility condition among those presented.”
- Throughout the results section the authors use “it was found”incorrectly.
We thank the Reviewer for the correction and apologise for the mistake. We corrected the text accordingly.
- Change has been showed or have showedto has been shown or have shown throughout the manuscript.
We thank the Reviewer for the suggestion, we modified along the text accordingly
- Consider breaking lines 573-581 into multiple sentences.
We thank the Reviewer for the suggestion, we modified the text accordingly:
“In the present study, 9 participants were pre-/peri-lingual deaf patients, analogous to the infants and toddlers included in the above mentioned article, while 11 were post-lingual deaf patients. Interestingly, a statistically significant higher gamma activity was found in the pre-/peri-lingual deaf group in the hemisphere contralateral to the CI side, which was negatively correlated with TAS-20 score and age in the general UCI sample, suggesting the occurrence of "facilitating EEG" activity in the processing of emotional stimuli. This could be possibly due to a longer wider compensatory plasticity mechanisms occurrence compared to the post-lingually deaf group, in order to achieve similar performance in emotion recognition skills [16].”
- The authors note a strong tendency for a negative correlation between right frontal gamma activity and correct responses but do not report the p-value. Including this information would aid in evaluating the significance of this statement.
We thank the Reviewer for the correction and apologise for the mistake. We corrected the text accordingly, specifying the p value: “p=0.053”.
- In the introduction, the authors mention differences in emotional processing between unilateral and bilateral CI users. It could be valuable to discuss how the results of this study might compare to those of bilateral CI users in the discussion section.
We thank the Reviewer for the suggestion. We do strongly think that such investigation would be a worthy research, therefore we added in the Discussion section, as possible future investigation, the following sentences in order to address the shortcoming:
“Moreover, it should be investigated the eventual influence of the bilateral CI condition in the evaluated phenomena, and in particular concerning the gamma activity patterns during emotion recognition tasks, given the already suggested differences between unilateral and bilateral CI users [11]. In fact, it could be expected a greater similarity concerning EEG patterns between bilateral CI users and NH controls than between unilateral CI users and NH group, as already suggested for instance for frontal alpha asymmetry patterns [12] and for theta and gamma neural correlates of working memory [13].”
- The conclusions could be more impactful by highlighting a main finding, rather than just stating that gamma activity is related to emotional music processing in adults with CIs.
We thank the Reviewer for the suggestion and modified the Conclusions section accordingly, as follows:
“EEG gamma activity appears to be fundamental to the processing of the emotional aspect of music and also to the psycho-cognitive emotion-related component in adults with CI. Although there wasn't a statistically significant difference in alexithymia scores between the UCI and NH groups, but emotion recognition performance was higher in NH compared to UCI participants, the neural correlates in the gamma band suggest the occurrence of compensatory neuroplasticity mechanisms trying to counteract sensory deprivation-induced deficits in emotion processing in the UCI group. In particular, relative higher gamma activity in the CI side corresponds to positive processes correlated with higher emotion recognition abilities, whereas gamma activity in the non-CI side may be related to positive processes inversely correlated with alexithymia and also inversely correlated with age. Therefore, gamma patterns seem to be a neurophysiological signature that accompanies the life of the CI patient from childhood to adulthood, apparently modifying itself, as suggested by the different results obtained in different studies”
- For Figure 1. It is unclear if “Emotion”is the title of the figure or the key. Consider moving it over the key or centering it over the graph for clarity.
We thank the Reviewer for the suggestion and modified the figure moving the word “Emotion” over the key.
- Line 490 fix {Citation}
We thank the reviewer for the correction and apologise for the error, which we have corrected as follows:
“The result of a deficit in emotion recognition in CI users in comparison to NH controls is a robust evidence, supported by many papers both in adults and children [17], [18], [19], [20], [21] population”

Reviewer 2 Report
Comments and Suggestions for Authors
First, I would like to thank you for reading and reviewing this manuscript. The study focuses on the emotional experiences of cochlear implant users, which I believe is an intriguing and practical topic. However, there are several issues that need to be addressed before considering this article for publication. Please refer to my specific recommendations below.
Keywords
Sensorineural hearing loss must be specified.
Introduction
It would be helpful to include basic information about sensorineural hearing loss, its main causes for both pre- and post-lingual cases, rehabilitation options, and when cochlear implantation is recommended. Additionally, pre- and postlingual types should be defined, as they are relevant to this investigation.
In the last paragraph of the introduction, 'categorization' is written in American spelling; however, other words are in British English. Please consistently use either American or British spelling and do not mix them.
Please provide an explanation of the abbreviation ’TAS-2’ at the end of this section.
Materials and methods
Considering the sample size of this investigation, it is certain that a limited number of participants were included, which may limit the results of this investigation. This must be included as a limitation at the end of the manuscript. Furthermore, since both pre- and postlingual cases were included, another possible significant influence on the results may be suspected, which should also be discussed.
When presenting the study and the control groups, basic demographic data (i.e., sex and age) would be beneficial in calculating whether the two groups were statistically significant, considering sex and age. This would ensure that the two groups are statistically comparable.
Furthermore, it would be helpful to provide details of the study population, such as the levels of hearing loss and the methods used for hearing testing, for the benefit of the readers.
Regarding the correct answers, did the right and left buttons display in a random order? It is crucial to prevent patients from manipulating the results.
At the end of the methods section, it is necessary to include a separate section on the statistical analysis. This should cover the statistical software used, including manufacturing details, normality tests, statistical tests used, and the significance level(s) needed to correctly interpret the results of this investigation.
Results
When presenting numbers, such as 67.708%, I believe it would be better to display them with two decimal places, for example, 67.71%.
‘Variable group’ must be clearly defined to avoid confusion for readers.
Figure 1. Please include a title of the y-axis.
TAS-20, first paragraph: It would be easier to comprehend the results if they were presented graphically, for example, by using boxplots.
‘..alexithymia occurrence among CI users on the base of aetiology’ Defining this aetiology as referring to pre- and postlingual cases would be more accurate. Furthermore, interpreting the results of the categorical analysis would be easier if this data were included in a table, also providing percentage values for each.
..without evidencing statistical significances..’ P-values must be presented to conclude this.
Last paragraph of this section: If we suspect a linear correlation analysis was obtained, the correlation between age and TAS-20 scores refers to higher scores with older ages. Please clarify this. Furthermore, if the ages of the basic demographic data were considered, an analysis of sex should also be performed.
Gamma asymmetry subsection
The term "ANOVA 2x2x3" is somewhat unusual; I suggest using "ANOVA test" or "Analysis of Variance" instead.
In this section, a few examples of higher gamma values were mentioned; however, it was not stated that these differences were statistically significant. Please correct each.
‘p<001’. This p-value must be corrected.
Figure 2. It must be clarified that the mean gamma activity values are presented on the y-axis.
‘(2)’. I was uncertain about the reference to this number; please clarify. If it pertains to article 2, then include it in the appropriate style.
‘..non-CI side higher values’. Please clarify that this a statistically significant difference.
When presenting the results of the correlation analysis, only the R-values were included. However, the p-values were not included, and it was not stated whether a significant/non-significant positive/negative correlation was observed. Please clarify this. Furthermore, a more detailed practical explanation of the results would be beneficial. This also applies to the other parts of the manuscript.
Figure 3. Specify gamma activity on the y-axis and provide the p-value for the Pearson correlation test. Additionally, explain abbreviations in the table caption.
Figure 5. It must be clarified that the mean (and SD) values are presented in this figure. However, from my perspective, a boxplot would bring more scientific value in this case.
Frontal gamma asymmetry
In the first sentence, please clarify that the frontal gamma asymmetries are considered in this case.
Please correct ‘wasn’t’ to was not.
‘residual hearing’ – Regarding this analysis, it is essential to specify whether the residual hearing levels or only residual hearing as a category was considered, as two different statistical methods are necessary for analysing each.
‘.. negative correlation between the mean gamma activity over the NON-CI side..’- Was this a significant negative correlation? The p-value should be provided.
Figure 6. The type of correlation should be specified, and the p-value should be included to draw a significant conclusion.
‘..evidencing higher gamma activity’ – It must be clarified that this was not a statistically significant difference.
Discussion
Firstly, the discussion should begin with a brief summary of the current study findings. Then, the authors can begin comparing the results with those of previous studies.
First sentence, [citation]. – Please remove this or include a reference article.
Lateralisation index – Once an abbreviation is introduced, it should be used exclusively throughout the entire text.
‘The important difference between the category of stimuli employed..’ – Could the authors specify this difference.
Generally, I do not find it necessary to include p-values and reference figures in the discussion since it should not replicate the results section.
At the end of the discussion, a separate section on this study's limitations must be provided, which should at least refer to the low number of participants and the fact that both pre- and postlingual cases were investigated.
I am looking forward to receiving the revised version of the manuscript
Comments on the Quality of English LanguageModerate editing is needed for this manuscript due to grammatical errors and unclear sentences. Consistent use of British/American spelling is required.
Author Response
REVIEWER 2
First, I would like to thank you for reading and reviewing this manuscript. The study focuses on the emotional experiences of cochlear implant users, which I believe is an intriguing and practical topic. However, there are several issues that need to be addressed before considering this article for publication. Please refer to my specific recommendations below.
Keywords
Sensorineural hearing loss must be specified.
We thank the Reviewer for the suggestion and we have modified accordingly.
Introduction
It would be helpful to include basic information about sensorineural hearing loss, its main causes for both pre- and post-lingual cases, rehabilitation options, and when cochlear implantation is recommended. Additionally, pre- and postlingual types should be defined, as they are relevant to this investigation.
We thank the Reviewer for the suggestion and we have added the following sentences in the Introduction section in order to address the comment (despite we understand that it is not a totally exhaustive description of the matter, that we feel goes beyond the scope of the present study):
“Sensorineural hearing loss can be caused by many different etiologies, and the assessment if the hearing loss is congenital or delayed and if its origin is genetic or nongenetic appear extremely relevant [22]. This condition corresponds to a wide range of congenital or acquired causes that may occur in the pre- peri- or post-lingual stages of language development. Many authors suggested that cochlear implants (CIs) represent a suitable treatment for children and adult patients with severe (hearing loss of 71–90 dB HL) to profound (hearing loss over 90 dB HL) in conversational frequency range (from 500 to 4000 Hz), where no or minimum benefit from a hearing aid after a trial period of 3–6 months was obtained, according to NICE Guidance on cochlear implantation (NICE 2009). For a more thorough description of the indications for CIs in accordance with age and hearing loss characteristics in both ears e.g. [23]. The causes of hearing loss in childhood (excluding infectious pathology of the middle ear) may be extrinsic (embryofoetopathy, meningitis, trauma, drug ototoxicity, noise trauma, etc.), genetic (e.g. alterations at the DFNB1 locus, STRC pathogenic variations or alterations at the DFN16 locus, SLC26A4 pathogenic variations, OTOF pathogenic variations, POU3F4 pathogenic variations or alterations at the DFNX2 locus) or both. The prevalence of genetic sensorineural hearing impairment is currently estimated to be 66% in industrialised countries and hereditary hearing loss can be divided into two broad categories: non-syndromic, estimated at 70-90%, and syndromic, estimated at 10-30% [24]. The time of deafness acquisition can be divided in: pre-lingual (onset of deafness before 2 years of age), peri-lingual (onset of deafness at 2–3 years of age), post-lingual (onset of deafness after 4 years of age).”
In the last paragraph of the introduction, 'categorization' is written in American spelling; however, other words are in British English. Please consistently use either American or British spelling and do not mix them.
We thank the Reviewer for the suggestion and we have modified accordingly in all the five cases where “categorization” was erroneously reported instead of “categorisation”.
Please provide an explanation of the abbreviation ’TAS-2’ at the end of this section.
We thank the Reviewer for the suggestion and apologise for the lack of clarity. “TAS-20” stands for “Toronto Alexithymia Scale (TAS-20 [25])”. The acronym was previously introduced in the Introduction section at line 113.
Materials and methods
Considering the sample size of this investigation, it is certain that a limited number of participants were included, which may limit the results of this investigation. This must be included as a limitation at the end of the manuscript. Furthermore, since both pre- and postlingual cases were included, another possible significant influence on the results may be suspected, which should also be discussed.
We thank the Reviewer for the suggestion and we have mentioned such limits in the Discussion section:
“The present study, as every research, of course present limitations, like for instance the limited number of participants and the analysis concerning pre-/peri-lingual and post-lingual deaf patients that would benefit from an enlargement of the sample, given the limited sample size tested. However, given the balanced number of CI users when considering pre-/peri-lingual etiology (n=9) and post-lingual etiology (n=11), it could be argued that the eventual influence of such factor would be mitigated in the present study. Future studies could explore these differences further. Moreover, it should be investigated the eventual influence of the bilateral CI condition in the evaluated phenomena, and in particular concerning the gamma activity patterns during emotion recognition tasks, given the already suggested differences between unilateral and bilateral CI users [11]. In fact, it could be expected a greater similarity concerning EEG patterns between bilateral CI users and NH controls than between unilateral CI users and NH group, as already suggested for instance for frontal alpha asymmetry patterns [12] and for theta and gamma neural correlates of working memory [13]. Finally, studies concerning emotion recognition and production in CI users appear particularly worthy for patients’ wellbeing, and to be taken into account for practical applications like rehabilitation programs, as very recently underlined [14].”
When presenting the study and the control groups, basic demographic data (i.e., sex and age) would be beneficial in calculating whether the two groups were statistically significant, considering sex and age. This would ensure that the two groups are statistically comparable.
We thank the Reviewer for the comment and apologise for the previous lack of information provided.
“NH and UCI groups were not statistically different concerning age (t=-1.357 p=0.186) and sex (Fisher's exact test two-tailed p=1)”
Furthermore, it would be helpful to provide details of the study population, such as the levels of hearing loss and the methods used for hearing testing, for the benefit of the readers.
We thank the Reviewer for the comment and apologise for the previous lack of clarity. Please consider the information we added in order to address the point:
“The audiometric inclusion criterion for UCI was a word comprehension rate of rate of at least 50% at 65 dB SPL [26], and this intensity was used for stimulus delivery in the experiment. The 50% threshold was set because a common measure of a listener's ability to understand speech in noise is the speech reception threshold (SRT) [27], which is defined as the SNR at which 50% of speech is correctly understood. Concerning the UCI group, the degree of hearing loss in the contralateral ear to the implanted one presented an average PTA (calculated including the frequencies 250-500-1000-2000-4000 Hz) ± standard deviation: 101.05 ± 13.77 dB.”
Regarding the correct answers, did the right and left buttons display in a random order? It is crucial to prevent patients from manipulating the results.
We thank the Reviewer for the comment and apologise for the previous lack of clarity. Yes, I confirm that half of the correct responses corresponded to the right button and half to the left button and that were displayed in a random order through the software E-Prime. According to the Reviewer’s comment we specified such detail in the protocol section, as follows:
“Half of the correct responses corresponded to the right button and half to the left button, displayed in a randomized order among participants.”
At the end of the methods section, it is necessary to include a separate section on the statistical analysis. This should cover the statistical software used, including manufacturing details, normality tests, statistical tests used, and the significance level(s) needed to correctly interpret the results of this investigation.
We thank the Reviewer for the comment and added the suggested section as follows:
2.6. Statistical Analysis
For the statistical analysis STATISTICA 12 software (StatSoft GmbH) was employed.
Significance level has been defined as α=0.05. Shapiro-Wilk test has been used in order to assess the normality of data distribution, that in a minority of cases did not fit the normal distribution, but it is worthy to note that, despite the statistical assumptions underlying the ANOVA methodology, it has been shown that ANOVA is not very sensitive to deviations from normality. In particular, simulation studies with non-normal distributions have shown that the false positive rate was not very affected by this violation of the normality assumption [28], [29], [30]. The UCI group was further divided into two groups: pre- and peri-lingual deaf CI users and post-lingual deaf CI users, in order to compare these two groups for TAS-20 scores, age and gamma power in the CI side and NON-CI side hemisphere.
ANOVA tests were performed in the comparisons between groups for the different EEG indices (for all groups: right and left average gamma power for each hemisphere, gamma LI calculated concerning each hemisphere and the right and left frontal area, frontal alpha asymmetry and average right and left frontal alpha power; for UCI: CI-side and NON-CI side average gamma power for each hemisphere, gamma LI calculated concerning each hemisphere and the CI-side and NON-CI side average gamma power over the frontal areas) and for behavioural/declared data (Correct Responses, TAS-20). Three factors were investigated: Group (two levels: UCI, NH), Emotion (two levels: Happy, Sad) and SNR (three levels: Quiet, SNR5, SNR10).
Unpaired t-tests were employed for the assessment of the differences between pre-/peri-lingual and post-lingual groups for the variables: age, correct responses, TAS-20 scores, average gamma power in the CI and NON-CI side over the hemispheres and over the frontal areas. Unpaired t-test was also performed comparing the left- and right-ear implanted patients in relation to the percentage of correct responses. Finally, t test was used for comparing the two groups, NH and UCI, concerning the variable age.
Fisher’s exact test two-tailed was used for testing the eventual statistically significant differences between the frequency of the groups concerning non-alexithymic/alexithymic traits occurrence in the UCI and NH group, and within the UCI group (comparing pre-/peri-lingual and post-lingual groups for the variables sex and non-alexithymic/alexithymic traits.
A logistic regression analysis between TAS-20 scores and sex for each group (UCI and NH) and subgroup (UCI: pre-/peri-lingual and post-lingual) was performed, in order to investigated the eventual relation between the mentioned continuous and dichotomic variables.
Pearson’s correlation analysis were performed between the EEG-based indices included in the study and some demographic, clinical and behavioural variables for both UCI and NH groups (for the NH group only the TAS-20 score, age and percentage of correct responses were included into the analysis).
Results
When presenting numbers, such as 67.708%, I believe it would be better to display them with two decimal places, for example, 67.71%.
We thank the Reviewer for the suggestion and modified accordingly.
‘Variable group’ must be clearly defined to avoid confusion for readers.
We thank the Reviewer for the suggestion, we defined the variable Group, in the Statistical description section. With that term we meant the variable “Group” with two levels: UCI and NH.
Figure 1. Please include a title of the y-axis.
We thank the Reviewer for the correction and apologise for the missing information. We have corrected the figure adding the title “Percentage of Correct Responses”.
TAS-20, first paragraph: It would be easier to comprehend the results if they were presented graphically, for example, by using boxplots.
We thank the Reviewer for the suggestion and added a new figure (Figure 2) concerning such results:
TAS-20
Figure 2. The boxplot represents the comparison between NH and UCI group for the Toronto Alexithymia Scale (TAS-20) scores, that did not report any statistically significant difference (t=-0.166 p=0.869).
‘..alexithymia occurrence among CI users on the base of aetiology’ Defining this aetiology as referring to pre- and postlingual cases would be more accurate. Furthermore, interpreting the results of the categorical analysis would be easier if this data were included in a table, also providing percentage values for each.
We thank the Reviewer for the suggestion and added a new table (Table 1) as follows:
Table 1. The table reports the frequencies of alexithymia in the pre-/peri-lingual deaf UCI participants and in the post-lingual deaf UCI participants. Fisher’s exact test did not show any statistical significance concerning such distribution (p=0.642).
..without evidencing statistical significances..’ P-values must be presented to conclude this.
We thank the Reviewer for the comment and apologise for the lack of clarity. We added the suggested details as follows:
“the time of CI use (TAS-20: r=0.130 p=0.583), the residual hearing in the contralateral ear (TAS-20: r=0.098 p=0.680), the duration of deafness (TAS-20: r=0.052 p=0.826) and the percentage of correct responses (TAS-20: r=-0.142 p=0.549).”
Last paragraph of this section: If we suspect a linear correlation analysis was obtained, the correlation between age and TAS-20 scores refers to higher scores with older ages. Please clarify this. Furthermore, if the ages of the basic demographic data were considered, an analysis of sex should also be performed.
We thank the Reviewer for the comments and apologise for the lack of clarity. Concerning the first point, we added the suggested details as follows (also p values for the mentioned correlations):
“Finally, for both UCI and NH groups it was performed an investigation of the correlation between TAS-20 score and age [31], [32], returning a correlation only for the NH group, that is higher TAS-20 scores were correlated with older ages (r=0.862 p=0.006), but not for the UCI one (r=0.127 p=0.593).”
Concerning the second point, we also performed a logistic regression analysis between TAS-20 scores and sex for each group, resulting in any statistical significance NH (Chi-square=0.281 df=1 p=0.596) UCI (Chi-square=2.053 df=1 p=0.152). Moreover, the same analysis was also performed on the UCI subgroups, not resulting in any statistical significance as well both for the pre-/peri-lingual group (Chi-square=1.189 df=1 p=0.275) and the post-lingual group (Chi-square=0.714 df=1 p=0.398). We added such further analysis in the text.
Gamma asymmetry subsection
The term "ANOVA 2x2x3" is somewhat unusual; I suggest using "ANOVA test" or "Analysis of Variance" instead.
We thank the Reviewer for the comment and replaced with “ANOVA test” throughout the text.
In this section, a few examples of higher gamma values were mentioned; however, it was not stated that these differences were statistically significant. Please correct each.
We added the expression “statistically significant” (marked in red in the revised manuscript) in order to highlight the reaching of the statistical significance, given the level of significance defined in the Statistical Method section (α=0.05). We also added p values throughout the text to all correlations reported, for which previously only r values have been specified.
‘p<001’. This p-value must be corrected.
We thank the Reviewer for the comment and we apologise for the typo, accordingly we replaced p<001 with p<0.001.
Figure 2. It must be clarified that the mean gamma activity values are presented on the y-axis.
We thank the Reviewer for the comment and apologise for the lack of clarity. Such information is specified in the figure’s caption:
“The graph represents the mean EEG gamma activity (PSD) estimated in the UCI group and resulting from the averaging of the signal acquired from the electrodes located in each hemisphere, as specified in the Methods section. Error bars stand for standard error. No statistically significant differences have been evidenced”
‘(2)’. I was uncertain about the reference to this number; please clarify. If it pertains to article 2, then include it in the appropriate style.
We thank the Reviewer for the comment, we referred to the formula number 2, reported in the Methods section. In order to be clearer to the reader we modified the sentence as follows:
“as defined by the formula (2) in the Methods section”
‘..non-CI side higher values’. Please clarify that this a statistically significant difference.
We thank the Reviewer for the comment, we clarified that we meant a statistically significant difference, given the level of significance defined in the Methods section (α=0.05).
When presenting the results of the correlation analysis, only the R-values were included. However, the p-values were not included, and it was not stated whether a significant/non-significant positive/negative correlation was observed. Please clarify this. Furthermore, a more detailed practical explanation of the results would be beneficial. This also applies to the other parts of the manuscript.
We thank the Reviewer for the comment and apologise for the lack of clarity. We added p values throughout the text to all correlations reported, for which previously only r values have been specified. Concerning the more practical explanation of results suggested by the Reviewer, we thought to keep such more “narrative” explanations in the Discussion section, in order to avoid a sort of preliminary discussion of results already in the Results section.
Figure 3. Specify gamma activity on the y-axis and provide the p-value for the Pearson correlation test. Additionally, explain abbreviations in the table caption.
We thank the Reviewer for the comment and apologise for the lack of clarity. We modified the caption accordingly to the suggestions, as follows:
“Figure 4. The scatterplot represents the correlation (Pearson’s r=0.456 p=0.043) between the mean EEG gamma activity reported in the hemisphere contralateral to the CI side and the alexithymia (Toronto Alexithymia Scale - TAS-20 scores) in the UCI group. PSD: power spectral density.”
Figure 5. It must be clarified that the mean (and SD) values are presented in this figure. However, from my perspective, a boxplot would bring more scientific value in this case.
We thank the Reviewer for the comment and apologise for the lack of clarity. We modified the caption accordingly to the suggestions, as follows:
“The graph represents the difference between Post-lingual and Pre-/Peri-lingual deaf UCI with respect to the mean gamma activity (PSD: power spectral density) in the non-CI side Hemisphere (p=0.045). Error bars stand for standard errors.”
We also modified the figure according to Reviewer’s suggestion, as follows:
Frontal gamma asymmetry
In the first sentence, please clarify that the frontal gamma asymmetries are considered in this case.
We thank the Reviewer for the comment and apologise for the lack of clarity. Accordingly we specified that detail.
Please correct ‘wasn’t’ to was not.
We thank the Reviewer for the correction and modified accordingly throughout the text.
‘residual hearing’ – Regarding this analysis, it is essential to specify whether the residual hearing levels or only residual hearing as a category was considered, as two different statistical methods are necessary for analysing each.
We thank the Reviewer for the comment and apologise for the lack of clarity. In order to address the comment, we specified in the text in the Methods section the following sentences:
“Concerning the UCI group, the degree of hearing loss in the contralateral ear to the implanted one presented an average PTA (calculated including the frequencies 250-500-1000-2000-4000 Hz) ± standard deviation: 101.05 ± 13.77 dB.”
‘.. negative correlation between the mean gamma activity over the NON-CI side..’- Was this a significant negative correlation? The p-value should be provided.
We thank the Reviewer for the comment and apologise for the lack of clarity. As mentioned in responses to previous comments, we added p values to all r values throughout the manuscript.
Figure 6. The type of correlation should be specified, and the p-value should be included to draw a significant conclusion.
We thank the Reviewer for the comment and modified the caption accordingly, as follows
“The scatter plot represents the statistically significant correlation (Pearson’s r=-0.446 p=0.049) between the average gamma activity (PSD: power spectral density) estimated in the frontal area contralateral to the CI side and the percentage of Correct Responses in the UCI group.”
‘..evidencing higher gamma activity’ – It must be clarified that this was not a statistically significant difference.
We thank the Reviewer for the comment and apologise for the lack of clarity. We modified the text accordingly, specifying that the comparison result did not reach the statistical significance.
Discussion
Firstly, the discussion should begin with a brief summary of the current study findings. Then, the authors can begin comparing the results with those of previous studies.
We thank the Reviewer for the suggestion and added the following sentences at the beginning of the Discussion section:
“Results showed no effect of the background noise, whilst support that gamma activity related to emotion processing in the UCI group presents alterations in comparison to the NH group, and that such alterations are also modulated by the deafness aetiology. In fact, main results showed that UCI group obtained poorer performances in emotion recognition than NH controls, and there was a negative correlation between TAS-20 scores and mean EEG gamma activity reported in the hemisphere contralateral to the CI side. Moreover, there was a correlation between the LI based on the CI side gamma activity values over the hemisphere and the percentage of correct responses. Finally there was a correlation between the average gamma activity estimated in the frontal area contralateral to the CI side and the percentage of correct responses in the UCI group.”
First sentence, [citation]. – Please remove this or include a reference article.
We thank the Reviewer for the correction and modified the text specifying the references we meant to mention, as follows:
“The result of a deficit in emotion recognition in CI users in comparison to NH controls is a robust evidence, supported by many papers both in adults and children [17], [18], [19], [20], [21] population.”
Lateralisation index – Once an abbreviation is introduced, it should be used exclusively throughout the entire text.
We thank the Reviewer for the correction and modified the text accordingly.
‘The important difference between the category of stimuli employed..’ – Could the authors specify this difference.
We thank the Reviewer for the comment and modified the sentence trying to better express ourself, as follows:
“The important difference between the category of stimuli employed in the present study and the one employed in the just mentioned research [33] is that musical stimuli are used here, whereas human non-verbal vocalisations (e.g. laughter, surprise) are used there.”
Generally, I do not find it necessary to include p-values and reference figures in the discussion since it should not replicate the results section.
We thank the Reviewer for the comment, however we do think that it could be of help for the reader to get also that cues in the Discussion, so we would prefer to keep those references.
At the end of the discussion, a separate section on this study's limitations must be provided, which should at least refer to the low number of participants and the fact that both pre- and postlingual cases were investigated.
We thank the Reviewer for the comment, please find the added section concerning limitations of the study:
“The present study, as every research, of course present limitations, like for instance the limited number of participants and the analysis concerning pre-/peri-lingual and post-lingual deaf patients that would benefit from an enlargement of the sample, given the limited sample size tested. However, given the balanced number of CI users when considering pre-/peri-lingual aetiology (n=9) and post-lingual aetiology (n=11), it could be argued that the eventual influence of such factor would be mitigated in the present study. Future studies could explore these differences further. Moreover, it should be investigated the eventual influence of the bilateral CI condition in the evaluated phenomena, and in particular concerning the gamma activity patterns during emotion recognition tasks, given the already suggested differences between unilateral and bilateral CI users [11]. In fact, it could be expected a greater similarity concerning EEG patterns between bilateral CI users and NH controls than between unilateral CI users and NH group, as already suggested for instance for frontal alpha asymmetry patterns [12] and for theta and gamma neural correlates of working memory [13]. Finally, studies concerning emotion recognition and production in CI users appear particularly worthy for patients’ wellbeing, and to be taken into account for practical applications like rehabilitation programs, as very recently underlined [14].”
I am looking forward to receiving the revised version of the manuscript
Comments on the Quality of English Language
Moderate editing is needed for this manuscript due to grammatical errors and unclear sentences. Consistent use of British/American spelling is required.
References:
[1] L. Bubbico, S. Ferlito, e G. Antonelli, «Hearing and Vision Screening Program for newborns in Italy», ANNALI DI IGIENE MEDICINA PREVENTIVA E DI COMUNITÀ, fasc. 5, 2021, doi: 10.7416/ai.2020.2401.
[2] P. Blamey et al., «Factors Affecting Auditory Performance of Postlinguistically Deaf Adults Using Cochlear Implants: An Update with 2251 Patients», Audiology and Neurotology, vol. 18, fasc. 1, pp. 36–47, ott. 2012, doi: 10.1159/000343189.
[3] B. M. S. Inguscio et al., «Gamma-Band Modulation in Parietal Area as the Electroencephalographic Signature for Performance in Auditory–Verbal Working Memory: An Exploratory Pilot Study in Hearing and Unilateral Cochlear Implant Children», Brain Sciences, vol. 12, fasc. 10, Art. fasc. 10, ott. 2022, doi: 10.3390/brainsci12101291.
[4] J. Campbell e A. Sharma, «Cross-Modal Re-Organization in Adults with Early Stage Hearing Loss», PLoS One, vol. 9, fasc. 2, Art. fasc. 2, feb. 2014, doi: 10.1371/journal.pone.0090594.
[5] P. Sandmann, K. Plotz, N. Hauthal, M. de Vos, R. Schönfeld, e S. Debener, «Rapid bilateral improvement in auditory cortex activity in postlingually deafened adults following cochlear implantation», Clin Neurophysiol, vol. 126, fasc. 3, Art. fasc. 3, mar. 2015, doi: 10.1016/j.clinph.2014.06.029.
[6] D. R. F. Irvine, «Plasticity in the auditory system», Hearing Research, ott. 2017, doi: 10.1016/j.heares.2017.10.011.
[7] C. M. McKay, «Brain Plasticity and Rehabilitation with a Cochlear Implant», Adv Otorhinolaryngol, vol. 81, pp. 57–65, 2018, doi: 10.1159/000485586.
[8] S. C. Purdy e A. S. Kelly, «Change in Speech Perception and Auditory Evoked Potentials over Time after Unilateral Cochlear Implantation in Postlingually Deaf Adults», Semin Hear, vol. 37, fasc. 1, Art. fasc. 1, feb. 2016, doi: 10.1055/s-0035-1570329.
[9] L. Lazzouni e F. Lepore, «Compensatory plasticity: time matters», Front Hum Neurosci, vol. 8, p. 340, 2014, doi: 10.3389/fnhum.2014.00340.
[10] D. A. Baldwin e L. J. Moses, «The Ontogeny of Social Information Gathering», Child Development, vol. 67, fasc. 5, Art. fasc. 5, ott. 1996, doi: 10.1111/j.1467-8624.1996.tb01835.x.
[11] S. Giannantonio, M. J. Polonenko, B. C. Papsin, G. Paludetti, e K. A. Gordon, «Experience Changes How Emotion in Music Is Judged: Evidence from Children Listening with Bilateral Cochlear Implants, Bimodal Devices, and Normal Hearing», PLoS One, vol. 10, fasc. 8, Art. fasc. 8, ago. 2015, doi: 10.1371/journal.pone.0136685.
[12] G. Cartocci et al., «Frontal brain asymmetries as effective parameters to assess the quality of audiovisual stimuli perception in adult and young cochlear implant users», Acta Otorhinolaryngol Ital, vol. 38, fasc. 4, pp. 346–360, ago. 2018, doi: 10.14639/0392-100X-1407.
[13] B. M. S. Inguscio et al., «Two are better than one: Differences in cortical EEG patterns during auditory and visual verbal working memory processing between Unilateral and Bilateral Cochlear Implanted children», Hearing Research, vol. 446, p. 109007, mag. 2024, doi: 10.1016/j.heares.2024.109007.
[14] O. Valentin, A. Lehmann, D. Nguyen, e S. Paquette, «Integrating Emotion Perception in Rehabilitation Programs for Cochlear Implant Users: A Call for a More Comprehensive Approach», J Speech Lang Hear Res, vol. 67, fasc. 5, pp. 1635–1642, mag. 2024, doi: 10.1044/2024_JSLHR-23-00660.
[15] M. Turrini, F. Cutugno, P. Maturi, S. Prosser, F. A. Leoni, e E. Arslan, «[Bisyllabic words for speech audiometry: a new italian material]», Acta Otorhinolaryngol Ital, vol. 13, fasc. 1, Art. fasc. 1, feb. 1993.
[16] H. Glick e A. Sharma, «Cross-modal plasticity in developmental and age-related hearing loss: Clinical implications», Hear Res, vol. 343, pp. 191–201, gen. 2017, doi: 10.1016/j.heares.2016.08.012.
[17] E. Ambert-Dahan, A.-L. Giraud, O. Sterkers, e S. Samson, «Judgment of musical emotions after cochlear implantation in adults with progressive deafness», Front Psychol, vol. 6, mar. 2015, doi: 10.3389/fpsyg.2015.00181.
[18] T. Hopyan, I. Peretz, L. P. Chan, B. C. Papsin, e K. A. Gordon, «Children Using Cochlear Implants Capitalize on Acoustical Hearing for Music Perception», Front Psychol, vol. 3, p. 425, ott. 2012, doi: 10.3389/fpsyg.2012.00425.
[19] S. Paquette, G. D. Ahmed, M. V. Goffi-Gomez, A. C. H. Hoshino, I. Peretz, e A. Lehmann, «Musical and vocal emotion perception for cochlear implants users», Hear Res, vol. 370, pp. 272–282, dic. 2018, doi: 10.1016/j.heares.2018.08.009.
[20] Y. Wang et al., «The Neural Processing of Vocal Emotion After Hearing Reconstruction in Prelingual Deaf Children: A Functional Near-Infrared Spectroscopy Brain Imaging Study», Front Neurosci, vol. 15, p. 705741, 2021, doi: 10.3389/fnins.2021.705741.
[21] C. M. Whipple, K. Gfeller, V. Driscoll, J. Oleson, e K. McGregor, «Do Communication Disorders Extend to Musical Messages?: An Answer from Children with Hearing Loss or Autism Spectrum Disorders», J Music Ther, vol. 52, fasc. 1, pp. 78–116, 2015, doi: 10.1093/jmt/thu039.
[22] M. M. Paparella, R. Y. Fox, e P. A. Schachern, «Diagnosis and Treatment of Sensorineural Hearing Loss in Children», Otolaryngologic Clinics of North America, vol. 22, fasc. 1, pp. 51–74, feb. 1989, doi: 10.1016/S0030-6665(20)31466-3.
[23] M. Manrique et al., «Guideline on Cochlear Implants», Acta Otorrinolaringologica (English Edition), vol. 70, fasc. 1, pp. 47–54, gen. 2019, doi: 10.1016/j.otoeng.2017.10.012.
[24] L. Jonard et al., «Genetic Evaluation of Prelingual Hearing Impairment: Recommendations of an European Network for Genetic Hearing Impairment», Audiology Research, vol. 13, fasc. 3, Art. fasc. 3, giu. 2023, doi: 10.3390/audiolres13030029.
[25] G. J. Taylor e R. M. Bagby, «New Trends in Alexithymia Research», Psychotherapy and Psychosomatics, vol. 73, fasc. 2, pp. 68–77, feb. 2004, doi: 10.1159/000075537.
[26] L. Bruns, D. Mürbe, e A. Hahne, «Understanding music with cochlear implants», Sci Rep, vol. 6, ago. 2016, doi: 10.1038/srep32026.
[27] H. Levitt, «Adaptive testing in audiology», Scand Audiol Suppl, fasc. 6, pp. 241–291, gen. 1978.
[28] G. V. Glass, P. D. Peckham, e J. R. Sanders, «Consequences of Failure to Meet Assumptions Underlying the Fixed Effects Analyses of Variance and Covariance», Review of Educational Research, vol. 42, fasc. 3, pp. 237–288, set. 1972, doi: 10.3102/00346543042003237.
[29] M. R. Harwell, E. N. Rubinstein, W. S. Hayes, e C. C. Olds, «Summarizing Monte Carlo Results in Methodological Research: The One- and Two-Factor Fixed Effects ANOVA Cases», Journal of Educational Statistics, vol. 17, fasc. 4, pp. 315–339, dic. 1992, doi: 10.3102/10769986017004315.
[30] L. M. Lix, J. C. Keselman, e H. J. Keselman, «Consequences of Assumption Violations Revisited: A Quantitative Review of Alternatives to the One-Way Analysis of Variance F Test», Review of Educational Research, vol. 66, fasc. 4, pp. 579–619, dic. 1996, doi: 10.3102/00346543066004579.
[31] O. R. Dobrushina et al., «Age-related changes of interoceptive brain networks: Implications for interoception and alexithymia», Emotion, apr. 2024, doi: 10.1037/emo0001366.
[32] A. K. Mattila, J. K. Salminen, T. Nummi, e M. Joukamaa, «Age is strongly associated with alexithymia in the general population», J Psychosom Res, vol. 61, fasc. 5, pp. 629–635, nov. 2006, doi: 10.1016/j.jpsychores.2006.04.013.
[33] G. Cartocci et al., «Higher Right Hemisphere Gamma Band Lateralization and Suggestion of a Sensitive Period for Vocal Auditory Emotional Stimuli Recognition in Unilateral Cochlear Implant Children: An EEG Study», Front Neurosci, vol. 15, p. 608156, 2021, doi: 10.3389/fnins.2021.608156.

Round 2
Reviewer 1 Report
Comments and Suggestions for Authors
The authors have done an excellent job addressing all the comments.
Comments on the Quality of English LanguageMinor editing is required. They are still missing words throughout the manuscript such as 'of' and 'the'
Author Response
We really thank the Reviewer for contributing to the improvement of the present manuscript, your work was truly deeply appreciated, thank you again!
We performed a further revision of the suggested minor grammatical mistakes and corrected them.
Reviewer 2 Report
Comments and Suggestions for Authors
Thank you for sending me the revised version of the manuscript. The authors have made significant improvements to the quality of the manuscript, and it now reads much better. Therefore, from my point of view, it can now be considered for publication.
Comments on the Quality of English LanguageThe overall text now reads better; however, some minor issues in English still can be found. Therefore, some further minor language editing would be beneficial.
Author Response

(The authors gave the same response as above.)
